# Machine learning-based predictive model for prevention of metabolic syndrome

**Hyunseok Shin**[1], **Simon Shim**[2], **Sejong Oh**[3]*

**1** Department of Computer Science, Dankook University, Youngin, South Korea, **2** Department of Applied Data Science, San José State University, San Jose, CA, United States of America, **3** Department of Software Science, Dankook University, Youngin, South Korea

* sejongoh@dankook.ac.kr

**Data Availability Statement:** Data cannot be shared publicly because of ethical restrictions. Data are available from the Korea Disease Control and Prevention Agency's Institutional Data Access / Ethics Committee (contact via division of Population Health Research [http://www.cdc.go.

## Abstract

Metabolic syndrome (MetS) is a chronic disease caused by obesity, high blood pressure, high blood sugar, and dyslipidemia and may lead to cardiovascular disease or type 2 diabetes. Therefore, the detection and prevention of MetS at an early stage are imperative. Individuals can detect MetS early and manage it effectively if they can easily monitor their health status in their daily lives. In this study, a predictive model for MetS was developed utilizing solely noninvasive information, thereby facilitating its practical application in real-world scenarios. The model's construction deliberately excluded three features requiring blood testing, specifically those for triglycerides, blood sugar, and HDL cholesterol. We used a large-scale Korean health examination dataset (n = 70, 370; the prevalence of MetS = 13.6%) to develop the predictive model. To obtain informative features, we developed three novel synthetic features from four basic information: waist circumference, systolic and diastolic blood pressure, and gender. We tested several classification algorithms and confirmed that the decision tree model is the most appropriate for the practical prediction of MetS. The proposed model achieved good performance, with an AUC of 0.889, a recall of 0.855, and a specificity of 0.773. It uses only four base features, which results in simplicity and easy interpretability of the model. In addition, we performed calibrations on the prediction probability and calibrated the model. Therefore, the proposed model can provide MetS diagnosis and risk prediction results. We also proposed a MetS risk map such that individuals could easily determine whether they had metabolic syndrome.

## Introduction

Metabolic syndrome (MetS) is a chronic disease caused by obesity, high blood pressure, hyperglycemia, and dyslipidemia [1]. Although there are slight differences in the details, there are five common risk factors: fasting plasma glucose, blood pressure, triglycerides, high-density lipoprotein cholesterol, and waist circumference. MetS is diagnosed if more than three factors among these are abnormal [1]. MetS has emerged as a major public health concern worldwide owing to the prevalence of MetS in adults in many urbanized countries steadily increasing to 20–30%. Furthermore, MetS increases the risk of cardiovascular disease and type 2 diabetes [2]. The prevalence of MetS in South Korea adults was reported to be 22.9% in 2018 [3].

kr]) for researchers who meet the criteria for access to confidential data.

**Funding:** This study was supported by the Ministry of Science, ICT (MSIT), Korea, under the High-Potential Individuals Global Training Program (2021-0-01531) and the R&D program of Development of AI ophthalmologic diagnosis and smart treatment platform based on big data(2018–0-00242) supervised by the Institute for Information & Communications Technology Planning & Evaluation (IITP). The funders had no role in study design, data collection and analysis, decision to publish, or preparation of the manuscript.

**Competing interests:** The authors have declared that no competing interests exist.

To improve this situation, noninvasive predictive studies have been conducted to easily detect and prevent MetS early. Noninvasive predictive models do not use invasive information obtained by penetrating the body or skin, such as blood tests, so continuous monitoring is possible at a simple, fast, and low cost. Noninvasive predictive studies have been published mainly in European and Asian countries [4–12]. Since 2015, many noninvasive studies have been conducted, and each study was conducted using samples of various nationalities, sizes, age groups, and prevalence (Table 1).

Most studies have been conducted on lifestyle-related and anthropometric features [4–7,9,11,13]. Gutiérrez-Esparza [13] attempted to find important features in lifestyle-related information. Gutiérrez-Esparza [13] viewed gender as an important factor and performed feature selection and model composition. However, in the final models, anthropometric features were evaluated as the main features, and although some lifestyle features were included in the final model, their roles were not significant [4,5]. The number of features used in the predictive models was between 4 and 17; more features tended to be used when lifestyle features were included, and the model became more complex.

The overall performance of the models was between 0.84 and 0.93 in terms of AUC, and most of them tended to have higher specificity than recall. Wang's study [6], which showed the best performance (AUC 0.93) using an artificial neural network, was characterized by the cumulative use of longitudinal data collected three times to increase performance. Fifteen features were used for prediction, including features of lifestyle and socioeconomic status, as well as physical features (waist circumference, age, and sex).

From an algorithmic point of view, interpretable models, such as decision trees (DTs) and logistic regression (LR), were half of the previous studies presented in Table 1, and the other half were hard-to-interpret black box models, such as ensembles, artificial neural networks, and random forests. Romero-Saldaña [9,10] constructed a simple rule-based decision tree using only the waist-to-height ratio and blood pressure. The AUC was not reported, and the specificity was quite high at 0.9 or higher, while the recall was low at 0.55 and 0.78. However, calibrations for the predictive probabilities were not evaluated, focusing only on classification performance. Datta [8] and Wang [6] performed calibrations, achieving good results in terms

**Table 1. Summary of previous studies on MetS prediction using noninvasive information.**

| Year | First author | Country | Sample size | Age (Mean) | MetS (%) | Classifier | Feature (Lifestyle) | AUC/Recall/Specificity/Calibration | | | |
|------|-------------|---------|-------------|-----------|----------|-----------|---------------------|-----------|-----------|-----------|-----------|
| 2008 | Kroon [12] | Netherlands | 642 | 23.1 | 7.5 | DT | 4(N) | - | - | - | N |
| 2015 | Hsiung [11] | Taiwan | 154 | 50.8 | 40.3 | LR | 3(N) | - | - | - | N |
| 2016 | Romero-Saldaña [10] | Spain | 1,185 | 45.1 | 14.9 | DT | 5(N) | - | 0.78 | 0.92 | N |
| 2018 | Romero-Saldaña [9] | Spain | 60,799 | 40.0 | 9.0 | DT | 5(N) | - | 0.55 | 0.95 | N |
| 2019 | Datta [8] | Germany | 2,314 | 55.2 | 40.7 | EN | 8≥(N) | 0.90 | - | - | Y |
| 2020 | Eyvazlou [7] | Iran | 468 | 40.3 | 37.6 | ANN | 17≥(Y) | - | 0.83 | 0.92 | N |
| 2020 | Wang [6] | Taiwan | 27,415 | - | 22.4 | ANN | 15 (Y) | 0.93 | 0.84 | 0.86 | Y |
| 2021 | Gutiérrez-Esparza [13] | Mexico | 2,289 | - | 50.1 | RF | 8~12(Y) | 0.84~ 0.88 | - | - | N |
| 2022 | Kim [5] | South Korea | 1,991 | 43.8 | 33.9 | RF | 14≥(Y) | 0.84 | 0.85 | 0.83 | N |
| 2022 | Xu [4] | China | 19,685 | 26.1 | 33.0 | LR | 8(N) | 0.86 | - | - | Y |

LR: Logistic regression; DT: Decision tree; EN: Ensemble; ANN: Artificial neural network; RF: Random forest; -: Unknown; Feature: Only unsynthesized features were counted in the final model. The feature is counted as one if a feature obtained by summing up the results of several questionnaires is included and then marked ≥; Lifestyle: Whether the final model includes lifestyle-related features, Yes/No; AUC: Area under the receiver operating characteristic curve; Calibration: Calibration performed or not, Yes/No.

of AUC. However, there is a limitation in that it is difficult to interpret predicted results due to the relatively large number of features and the high complexity of the model.

We can achieve satisfactory accuracy when we apply predictive models from previous studies to real life, but they require many features (information) for prediction and do not provide predictive probabilities. However, some models are difficult to interpret. A practical MetS predictive model should achieve satisfactory accuracy with minimal features and explain the prediction results such that they are understandable. Furthermore, if a predictive model predicts both the presence or absence of a disease and the risk probability, it will be more helpful for understanding health status.

In this study, we developed a practical predictive model to help prevent MetS. First, we explored the most informative features to obtain a sufficient predictive performance. We developed novel synthetic features for candidate features and performed feature selection. Second, we focus on tree-based classification algorithms. We constructed models from the basic DT (CART) to ensembles (Random Forest, Extreme Gradient Boosting) and deep learning-based trees (TabNet) and compared their performance by AUC, sensitivity, specificity, balanced accuracy, and a number of features. Third, we propose a MetS management tool that visually constructs a predictive model. The outcome of the tree model is expressed in a decision structure that has the advantage of high interpretability. We propose a visual tool (MetS risk map) for MetS prevention by reconstructing decision structures in a more user-friendly form and adding risk probabilities.

## Materials and methods

### Procedure

Fig 1 depicts the process used to develop our MetS machine learning predictive model. We used the health checkup records to develop the model. These records contained the core elements of anthropometry and blood test results that could identify MetS. The data included survey results on lifestyle, diet, family history, and medical history. We extracted as many features (data attributes) as possible from these data to discover informative features for diagnosis and investigated previously known indicators. We also synthesized new anthropometric features using waist circumference, blood pressure, and diagnostic criteria and created new dietary-related features by borrowing the evaluation items of the Korean healthy eating index [14] and inflammatory index [15]. After excluding subjects with outliers and missing values, all features were arranged in a tabular data set.

In this tabular dataset, we divided 10% of the data into a test dataset for performance evaluation. The rest were used as a training dataset for model learning, and 10% of it was re-divided into a validation dataset, which was repeated 30 times to enable a stable performance

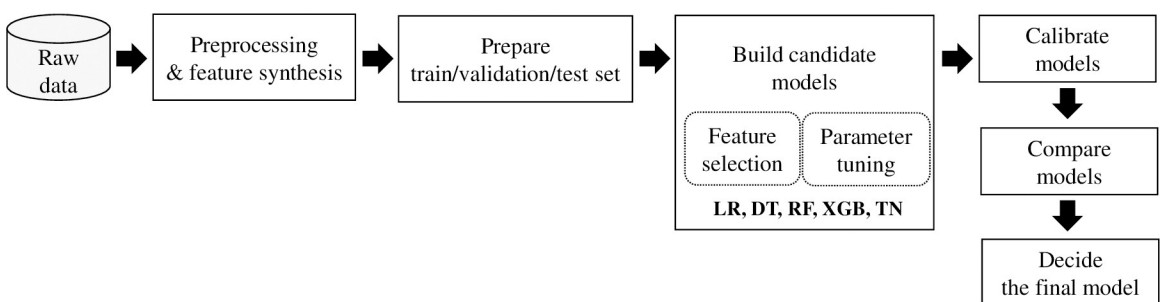

**Fig 1. Overall model development procedure.**

comparison when finding the best learning model. Except for the validation and test datasets, only the training dataset was undersampled to adjust the ratio of cases with and without MetS to 1:1.

Using the training/validation sets, we built five machine learning models based on LR, CART (DT), Random Forest (RF), Extreme Gradient Boosting (XGB), and TabNet (TN). Feature selection was performed for the five models to determine the optimal features. After the feature selection, we performed parameter tuning for the five models to show as high an AUC as possible. Finally, we evaluated the prediction accuracy of the models using a test dataset. We also evaluated the practicality of each learning model based on performance metrics, such as recall, specificity, and balanced accuracy, as well as the calibration plot and Brier score, number of features, and interpretability.

Data preprocessing was implemented by R(4.0.4) and tidyverse package(1.3.1), and learning model development and model evaluation were implemented by Python(3.8.13), PyTorch (1.12.1), Scikit-learn(1.1.3), XGBoost(1.6.2), and TabNet(4.0).

## Raw data

This study was based on health checkup records collected from South Korea: eight major metropolitan cities (Seoul, Incheon, Daegu, Busan, Gwangju, Ulsan, Daejeon, and Sejong) and eight other provinces (Gyeonggi-do, Gangwon-do, Gyeongsangnam-do, Chungcheongbuk-do, Chungcheongnam-do, Jeollanam-do, Jeollabuk-do, and Jeju-do). The records were obtained from the Korea Genome and Epidemiology Study conducted by the Korea Disease Control and Prevention Agency [16]. The survey collected lifestyle, medical history, dietary habits, food intake, and anthropometric and clinical measurements to identify risk factors for chronic diseases common to Koreans. They include all the factors necessary for diagnosing MetS: waist circumference, systolic and diastolic blood pressure, fasting glucose, triglycerides, and HDL cholesterol. The survey was conducted from 2004 to 2013, and records of 173,209 adults aged 40 years or older were collected.

A total of 70,370 participants were selected from the source records and used according to the following criteria: 1) subjects aged < 70 years; 2) exclusion of subjects with the following diseases that can affect dietary habits: hypertension, diabetes, hyperlipidemia, stroke, fatty liver, angina pectoris, thyroid disease, and cancer; 3) exclusion of subjects with missing values and outliers regarding diet, blood tests, and anthropometric measurements. Table 2 summarizes the characteristics of the selected participants, focusing on the MetS factors. The

**Table 2. The characteristics of the selected participants (n = 70,370).**

| | All (n = 70,370) | Without MetS (n = 60,775) | With MetS (n = 9,595) |
|---|---|---|---|
| Age(year) | 50.7 ± 7.6 | 50.3 ± 7.5 | 52.8 ± 7.9 |
| Sex | | | |
| male, n(%) | 21,652 (31%) | 17,424 | 4,228 |
| female, n(%) | 48,718 (69%) | 43,351 | 5,367 |
| Systolic blood pressure (mmHg) | 120.4 ± 14.3 | 118.5 ± 13.3 | 132.5 ± 14.4 |
| Diastolic blood pressure (mmHg) | 75 ± 9.7 | 73.9 ± 9.2 | 82.3 ± 9.7 |
| Waist circumference (cm) | 79.6 ± 8.4 | 78.3 ± 7.7 | 87.9 ± 7.4 |
| Fasting blood glucose (ml/dl) | 91.4 ± 14.7 | 89.6 ± 11.8 | 102.7 ± 23.5 |
| HDL cholesterol (mg/dl) | 55.2 ± 13 | 56.9 ± 12.7 | 44.6 ± 9.4 |
| Triglycerides (mg/dl) | 117.3 ± 83 | 102.4 ± 61.9 | 211.7 ± 126.1 |
| Number of risk factors | 1.2 ± 1.1 | 0.8 ± 0.8 | 3.3 ± 0.6 |

Institutional Review Board (IRB) of Dankook University granted approval for the study protocol and waived the requirement for obtaining informed consent from participants (DKU 2021-06-008).

## Preprocessing and feature synthesis

We used the presence or absence of MetS as a class label and defined MetS based on the criteria proposed in 2005 by the revised National Cholesterol Education Program-Adult Treatment Panel III (revised NCE APT III) [1]. Waist circumference for abdominal obesity followed the criteria suggested by the Korean Society for Obesity [17], which is recommended for Koreans. In summary, the diagnostic criteria for MetS in this study were as follows: 1) increased waist circumference ($\geq$90 *cm* for males and $\geq$85 *cm* for females); 2) elevated blood pressure (systolic blood pressure $\geq$130 *mmHg* or diastolic blood pressure $\geq$85 *mmHg*); 3) elevated fasting blood glucose ($\geq$100 *mg/dl*); 4) elevated triglycerides ($\geq$150 *mg/dl*); 5) reduced HDL cholesterol ($\leq$40 *mg/dl* for male and $\leq$50 *mg/dl* for female). A state that exceeded the standard was defined as having risk factors, and a participant with three or more of the following risk factors was diagnosed with MetS.

We extracted as many features (predictors) as possible to identify informative features for diagnosing MetS. We targeted only noninvasive measurement items, excluding blood test items. Table 3 summarizes the features extracted from the health checkup records. In total, 237 features were extracted and classified into three types according to their attributes: anthropometric, survey-based, and synthesized. Anthropometric features consist of body information measured by professional examination institutions and body shape-related features synthesized using this information. These synthesized features are anthropometric indices that describe body fat distribution [4]. Survey-based features contain lifestyle-related information, such as food intake, drinking, smoking, and exercise, and were collected through questionnaires [16].

In addition to these two types of features, we synthesized new features. Waist circumference and blood pressure, which are noninvasive information and risk factors for MetS, were synthesized as follows:

$$f(z) = elliot\ sigmoid\ (z) = \frac{0.5 \times z}{1 + |z|} + 0.5,$$

$$where\ z = \frac{x - c}{r \times c}, \quad c = diagnositc\ criteria, \quad r = 0.1$$

**Table 3. List of features extracted from the health checkup records.**

| Type (N) | Category (N) | Feature | Synthetic |
|---|---|---|---|
| Anthropometric (19) | Basic (9) | sex, age, height, weight, waist circumference, hip circumference, pulse, systolic blood pressure, diastolic blood pressure | No |
| | Body shape (10) | BMI, BFP, WHR, WHtR, BRI, ABSI, CUN-BAE, C-INDEX, AVI, BAI | Yes |
| Survey-based (151) | Lifestyle (17) | habit and amount: exercise, sleep, smoking, drinking, eating | No |
| | Diet (129) | intake for 106 foods, calories, 22 nutrients | No |
| | Family history (5) | hypertension, diabetes, angina, stroke, cancer | No |
| Synthesized (67) | Body-based (9) | SBP, DBP, BP, WC, bWC, BPWC_con, BPWC_mul, BPWC_add, BPWC_diff | Yes |
| | Diet -based (58) | Korean health eating index items, dietary inflammatory index items | Yes |

N: The number of features.

**Table 4. List of types and details of synthetic features used in this study.**

| Type | Feature | Description | Formula |
|---|---|---|---|
| Anthropometric (Known) | BMI | Body Mass Index | weight (kg)/height (m)$^2$ |
| | BFP | Body Fat Percentage | Male: $0.567 \times$ waist (cm) $+ 0.101 \times$ age $- 31.8$<br>Female: $0.439 \times$ waist (cm) $+ 0.221 \times$ age $- 9.4$ |
| | WHR | Waist to Hip Ratio | waist (cm)/hip (cm) |
| | WHtR | Waist-to-Height Ratio | waist (cm)/height (cm) |
| | BRI | Body Roundness Index | $364.2 - 365.5 \times \sqrt{1 - (\text{waist (cm)}/\pi \times \text{height (cm)})^2}$ |
| | ABSI | A Body Shape Index | $\text{waist (m)}/\left( \text{BMI}^{2/3} \times \text{height(m)}^{1/2} \right)$ |
| | CUN-BAE | Clinica Universidad de Navarra-Body Adiposity Estimator index | $-44.988 + (0.503 \times age) + (10.689 \times sex) + (3.172 \times BMI) - (0.026 \times BMI^2) + (0.181 \times BMI \times sex) - (0.02 \times BMI \times age) - (0.005 \times BMI^2 \times sex) + (0.00021 \times BMI^2 \times age)$ |
| | C-INDEX | Conicity Index | $\text{waist (m)}/(0.019 \times \sqrt{\text{weight(kg)}/\text{height(m)}})$ |
| | AVI | Abdominal Volume Index | $\{2 \times \text{waist(cm)}^2 + 0.7 \times (\text{waist(cm)} - \text{hip(cm)})^2\}/1000$ |
| | BAI | Body Adiposity Index | $100 \times hip\ (m)/\text{height}(m)^{3/2} - 18$ |
| Synthetic (developed by us) | SBP | Risk based on sbp (Scaled sbp) | $0.5 \times \left\{ \frac{sbp-130}{4.5} \right\}/\left\{ 1 + \text{abs}\left( \frac{sbp-130}{4.5} \right) \right\} + 0.5$ |
| | DBP | Risk based on dbp (Scaled dbp) | $0.5 \times \left\{ \frac{dbp-85}{4.5} \right\}/\left\{ 1 + \text{abs}\left( \frac{dbp-85}{4.5} \right) \right\} + 0.5$ |
| | BP | Risk based on bp (Scaled bp on the higher side) | Max(SBP,DBP) |
| | WC | Risk based on waist (Scaled Waist) | Male: $0.5 \times \left\{ \frac{waist-90}{9} \right\}/\left\{ 1 + \text{abs}\left( \frac{waist-90}{9} \right) \right\} + 0.5$<br>Female: $0.5 \times \left\{ \frac{waist-85}{8.5} \right\}/\left\{ 1 + \text{abs}\left( \frac{waist-85}{8.5} \right) \right\} + 0.5$ |
| | bWC | BMI and WC interaction | BMI×WC |
| | BPWC_add | Risk based on BP and WC | BP+WC |
| | BPWC_mul | BP and WC interaction | BP×WC |
| | BPWC_diff | Imbalance between BP and WC | BP−WC |
| | BPWC_con | Interaction between BP and WC above certain values | Max(BP×WC−0.25, 0) |

waist = waist circumference; hip = hip circumference; sbp = systolic blood pressure; dbp = diastolic blood pressure; bp = blood pressure. SBP and DBP in capital letters are synthetic features that differ from those of sbp and dbp.

The measured $x$ of waist circumference or blood pressure was scaled based on each diagnostic criterion and then applied to the sigmoid series function. In the case of blood pressure, $c$ in the denominator was substituted with 45, which is the difference between systolic and diastolic blood pressure. In addition, the final synthetic feature of blood pressure was higher after obtaining systolic and diastolic synthetic features. WC and BP, which are the basis materials for the other synthetic features, have the following properties: 1) this value ranges between 0 and 1; 2) when this value is 0.5, the original value is the same as the diagnostic criterion; 3) as this value approaches 1, it significantly exceeds the diagnostic criterion; 4) this value more sensitively reflects changes near the diagnostic criterion. Table 4 summarizes the details of the features, and further details can be found in our previous study [18].

Moreover, additional features were created by synthesizing these basic unit features (BP and WC) again and calculating lifestyle-related features according to each measurement item of the Korean health eating index [14] and the dietary inflammatory index [15] (Tables 3 and 4). Synthetic features are marked "Yes" in the Synthetic column in Table 3, and their names are capitalized to distinguish them from the raw features. In this study, "raw features" refer to features extracted from a single piece of information, such as height, weight, and waist circumference, while synthetic features refer to new features made using two or more raw features.

## Prepare train/validation/test sets

We divided the dataset into training and test datasets at a ratio of 9:1. We re-split the training dataset at a 9:1 ratio and used a small portion as a validation dataset. Because there was an imbalance between classes, the prevalence of MetS was 13.6%, and the ratio of MetS to nonmetabolic cases was adjusted to 1:1 using the undersampling method. Undersampling was performed only on the training dataset, and the original MetS prevalence was maintained in the validation and test datasets. The process of separating the validation dataset from the training dataset was repeated 30 times to construct a dataset containing various possible non-MetS cases (Table 5).

## Build candidate models

**Feature selection.**   In the candidate model-building phase, feature selection and parameter tuning were performed for the five classification algorithms. We attempted to screen no more than 10 final features for the practical use of diagnostic models. Among the three types of features classified in Table 3, the most informative features were selected after three rounds, as shown in Fig 2. Each round also performs a feature selection process, as shown in Fig 3.

First, three rounds followed the following procedure: Round 1 selects no more than 10 best features from each of the anthropometric and survey-based features. Round 2 combines each selected feature and then selects no more than 10 best features from the feature set. Finally, Round 3 selects no more than 10 best features from the feature set that combines the features selected in Round 2 with our proposed synthetic features.

Each step followed the following procedure: First, we evaluated the feature importance and selected the top 30 or fewer features. Feature importance was evaluated using the method provided by each classifier, and LR was based on the exponential conversion of the coefficients. In the second step, we selected up to 10 features that we had previously obtained using the recursive feature elimination (RFE) method. The RFE is a wrapper feature selection method, which is a model-dependent method based on the evaluation of the learning model used. In this process, we used the RFE and RFECV functions of sklearn and the AUC as the evaluation criteria. In the third step, we selected the best performance feature set from all 10 possible feature combinations. We constructed the model using each feature combination and selected the feature set of the models with the highest AUC.

**Classification algorithms.**   The selection criteria for the five models are delineated as follows: First, LR was chosen as a benchmark model for performance comparison. LR, a conventional algorithm alongside DT, is renowned for its interpretability. DT and TN were chosen based on their interpretive characteristics [19–21]. DT possesses an innate quality of being easily comprehensible by non-experts if it has a reasonable number of nodes. TN, a recent deep learning model, can generate feature maps through a learnable mask, representing features highly correlated with prediction results. It was postulated that TN's interpretability could render it the most appropriate option if it exhibits a discernable difference in performance compared to existing models. RF and XG were selected for performance evaluation because the

**Table 5. Characteristics of the training/validation/test datasets.**

| Sample | Number | Size | MetS (%) |
|---|---|---|---|
| All | 1 | 70,370 | 13.6 |
| Training | 30 | 15,560 ± 46 | 50.0 |
| Validation | 30 | 6,334 | 13.7 ± 0.4 |
| Test | 1 | 7,037 | 13.5 |

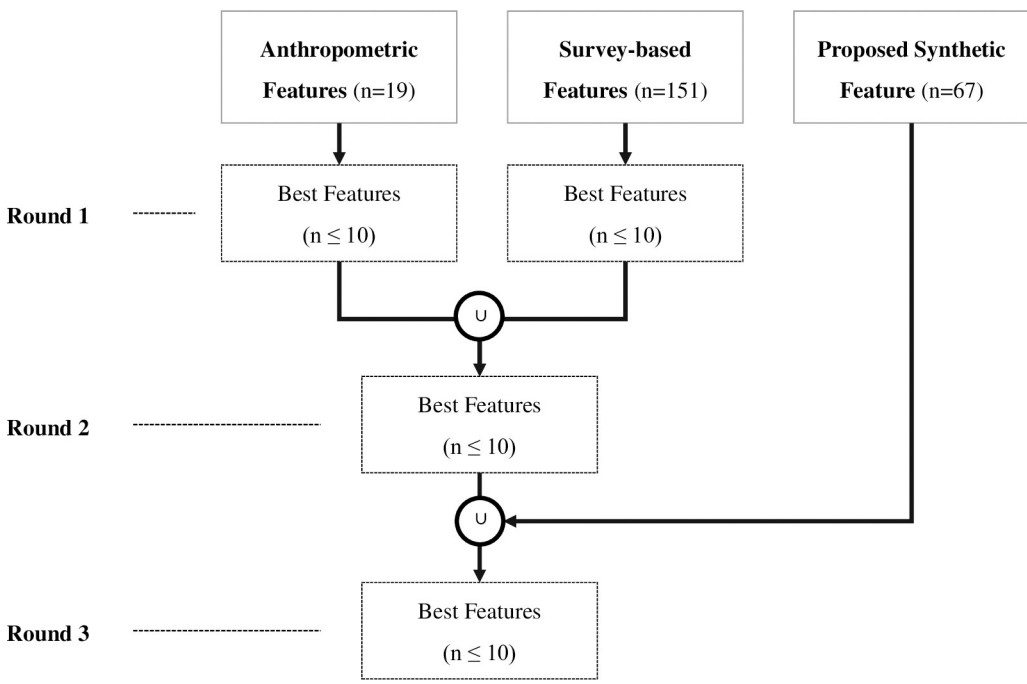

**Fig 2. Diagram for feature selection process.**

ensemble model RF and boosting family XG are known to be the highest performing models in tabular data [22–24]. Consequently, the performance of these models served as an upper limit to gauge the positioning of interpretable models DT and TN.

The rationale for selecting the DT-series algorithm lies in its inherent flexibility, which can be attributed to two primary characteristics [25]. First, decision trees are categorized as non-parametric methods, thereby implying that they are not constrained by any assumptions pertaining to the distribution of the space. Second, decision trees are distance-based models that neither require normalization nor scale conversion and are robust to the presence of outliers.

TN is a novel deep neural network (DNN) architecture that utilizes a decision-tree-based approach to handle tabular data [21]. TN is capable of 1) processing raw data without any pre-processing, 2) selecting features in an instance-wise manner using sequential attention, and 3) mimicking an ensemble by sequentially repeating DNN blocks called "Steps" [21]. The key element of TN is the learnable mask used for feature selection, which enables the implementation of output manifolds similar to those of Decision Trees. The TN architecture is built by repeating the Step building block, where each Step receives attention information from the previous Step, learns the mask, selects the features, and outputs the results.

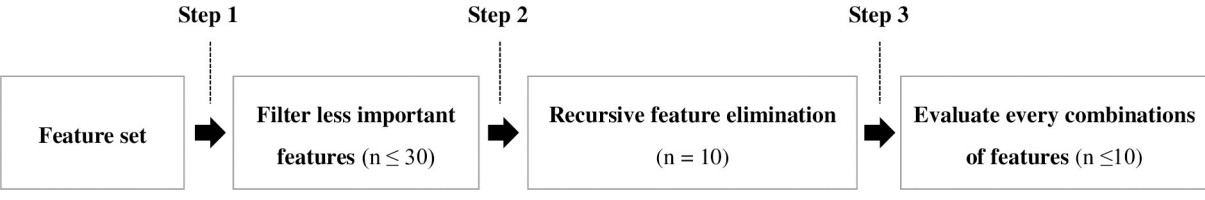

**Fig 3. Feature selection process for each round.**

**Parameter tuning.** Parameters were tuned using the ParameterGrid function of the Model Selection class provided by sklearn. First, the parameters were preset for each classifier according to the preset parameter column in Table 6 to generate the basic models for parameter tuning. All other non-predefined parameters used default values. Then, as shown in the grid parameter column in Table 6, the optimal combination was determined by changing the values of the main parameters for each classifier. A grid search was conducted using AUC as the evaluation criterion.

## Model calibration

Calibration is an essential component of predictive model evaluation for medical decision-making, diagnosis, and prognosis [26]. Calibration is a measurement of how well the predicted probability of an event matches the true underlying probability of the event [27]. In practice, a good calibration means that a predicted probability of 0.9 actually occurs with a probability of 0.9 [27]. Clinically, the probability of occurrence can also be interpreted as a risk that has practical significance. In the decision-making process, it is more useful to refer to continuous values for risks, such as probability, rather than simply broad classifications, such as MetS [27]. Therefore, it is important that our models are well-calibrated and have good discrimination.

We used three methods to calibrate the predictive probability of a diagnosis model: Platt scaling, isotonic regression, and Pozzolo's calibration [28,29]. These methods are designed for binary classification and require the use of an independent calibration set to obtain good calibration probabilities [29].

Platt scaling is the most effective method for calibrating SVM prediction probabilities when the predicted probabilities are distorted in a sigmoid shape [29]. The calibrated probability is obtained by passing the output f(x) of the diagnostic model through the sigmoid function:

$$P(y = 1|f) = \frac{1}{1 + \exp(Af + B)}$$

**Table 6. Parameter tuning settings by classifier.**

| Classifier | Preset parameters | Grid of parameters |
|---|---|---|
| LR | random_state: 100,<br>penalty: 'none' | solver: ['newton-cg','lbfgs','sag','saga'] |
| DT<br>(CART) | random_state: 100 | criterion: ['gini', 'entropy'],<br>max_depth: [2, 3, 4, 5],<br>min_samples_split: [0.01, 0.1, 0.2],<br>max_features: [None, 'log2', 'sqrt'],<br>splitter: ['best', 'random'],<br>min_samples_leaf: [50, 100, 200] |
| RF | random_state: 100 | bootstrap: [True, False],<br>max_depth: [3, 4, 5, None],<br>min_samples_leaf: [2, 4],<br>min_samples_split: [2, 5, 10],<br>n_estimators: [200, 500, 1000, 2000] |
| XGB | random_state: 100,<br>objective: 'binary:logistic',<br>booster: 'gbtree',<br>n_jobs: -1,<br>seed: None | n_estimators:[200,500,1000],<br>max_depth: [4,6,8],<br>learning_rate: [0.01, 0.05, 0.1],<br>subsample: [0.6, 0.7, 0.8] |
| TN | seed: 100<br>optimizer_fn:torch.optim.Adam,<br>optimizer_params: dict(lr = 2e-2),<br>scheduler_params:{"step_size":50, "gamma":0.9},<br>scheduler_fn:torch.optim.lr_scheduler.StepLR,<br>mask_type:'entmax', | n_steps: [2, 3],<br>gamma: [0.8, 1, 1.2],<br>n_independent: [2,3,4],<br>n_shared: [2,3,4],<br>momentum: [0.01,0.02,0.03], |

Parameters A and B are fitted using the maximum likelihood estimation method from the fitting set, and gradient descent is used to find the following solution [29]:

$$\underset{A,B}{\operatorname{argmin}}\{-\sum_i y_i \log(p_i) + (1 - y_i)\log(1 - p_i)\}, \ \text{where } p_i = \frac{1}{1 + \exp(Af_i + B)}$$

Isotonic regression is a more generalized method, with the only restriction being that the mapping function is monotonically increasing (isotonic) [29]. The basic assumption in isotonic is:

$$y_i = m(f_i) + \epsilon_i,$$

where $m$ is an isotonic function, $f_i$ is the prediction from the model, and $y_i$ is the class label. Then, given a fitting set $(f_i, y_i)$, the isotonic regression problem is to find the isotonic function $\hat{m}$ such that

$$\hat{m} = argmin_z \sum (y_i - z(f_i))^2$$

Isotonic regression has the advantage of being able to calibrate any monotonous distortion well, whereas overfitting is likely to occur when data are scarce [29].

Pozzolo's method corrects the predictive probability of the undersampled model [28]. Undersampling resulted in a mismatch in the distribution between the training and test sets. In other words, the learning model was based on the distribution of the training set, but the test set used in the evaluation was similar to the distribution before undersampling. Therefore, it is necessary to adjust the bias caused by the difference between these two distributions in the predictive probabilities of the learning model [28]. The bias-corrected probability $p'$ is obtained using the following equation:

$$p' = \frac{\beta p_s}{\beta p_s - p_s + 1},$$

where $\beta$ is the probability of selecting an undersampled negative instance from all negative instances and $p_s$ is the predictive probability of a model trained on undersampled datasets. The advantage of Pozzolo's method is that it is not only possible to calculate the optimal threshold in a simple way based on mathematical theory, but it also does not require additional fitting sets for calibration. The optimal thread hold equals the probability of selecting a positive from the entire dataset.

Calibration is typically measured as a set of predictions and not as an individual prediction. It is impossible to directly measure the true underlying probability of a one-time event because only one event occurs or does not occur [27]. The Brier score is a typical method and is the mean squared error for a set of predictions between the actual and predicted probabilities [26]. Given a set of predictions $\hat{p}$ with true probabilities $p$, the Brier score is

$$\frac{1}{n}\sum_{i=1}^{n}(p_i - \hat{p}_i)^2$$

A lower score indicates better accuracy, but no "good" criterion has been established [27]. Therefore, another measurement is required to determine whether the calibration is significant, such as the Spiegelhalter z-test [26]. This method presents a criterion for determining the

significance of calibration by decomposing the Brier score. Spiegelhalter's z-test is defined as

$$z = \frac{\sum_{i=1}^{n}(y_i - \hat{p}_i)(1 - 2\hat{p}_i)}{\sqrt{\sum_{i=1}^{n}(1 - 2\hat{p}_i)^2 \hat{p}_i(1 - \hat{p}_i)}},$$

where $y_i$ is the $i$th true class label and $\hat{p}_i$ is the $i$th predicted probability. Statistically significant scores (i.e., z<-1.96 or z > 1.96) generally indicate poor calibration because the values of z follow an asymptotically standard normal distribution, and the null hypothesis is that the model is well-calibrated [27].

There are ways to evaluate calibrations using a graphical approach to compensate for the limitations of summary statistics, such as the Brier score and Spiegelhalter's z-test statistic. A calibration plot, also called a reliability plot, is a graph that connects the corresponding points with the prediction probability on the x-axis and the actual probability on the y-axis. The plot includes a diagonal line that is fully calibrated. The advantage of the calibration plot is that miscalibration patterns can be easily identified [27].

## Model comparison metrics

In medical research, the AUC is widely used for discriminant evaluation [26,27,30]. The ROC plot depicts the trade-off between recall and specificity. In the plot, the x-axis denotes recall, y-axis the specificity, and AUC the area under the ROC curve.

Recall and specificity are two components that measure the validity of diagnostic models with dichotomous predictions [30]. Comparing the predicted diagnosis with the actual health status, it was divided into four cases: True positive (TP), False positive (FP), True negative (TN), and False negative (FN). TP is a case in which a patient with the disease is predicted to be positive. FP is a case in which a patient without a disease is predicted to be positive. TN occurs when a patient without a disease is predicted to be negative. FN is a case in which a patient with a disease is predicted to be negative. The recall of a diagnosis refers to the ability of the model to correctly identify patients with the disease, whereas the specificity of a diagnosis refers to the ability of the model to correctly identify patients without the disease:

$$recall = True\ positives/(True\ positives + False\ negatives)$$

$$specificity = True\ negatives/(True\ negatives + False\ positives)$$

The AUC also has robust properties in terms of prevalence because recall and specificity are not affected by the prevalence of the disease [30]. In addition, the AUC can have values between 0 and 1 because the two axes of ROC are recall and specificity with values between 0 and 1; the AUC of the pure random model is 0.5, and the AUC of the perfect model is 1 [27]. We also used balanced acuity to evaluate discrimination, as it is a robust indicator of prevalence.

$$balanced\ accuracy = (recall + specificity)/2$$

## Results

### Feature selection

**Selected features and their importance.** Seventeen features were selected from the five classifiers, as listed in Table 7. There were eight anthropometric features, six of which were our

**Table 7. Final selected features.**

| Classifier | Final selected features (N) | | Raw features (N) | |
|---|---|---|---|---|
| LR | *WC, *BP, CUN-BAE, carbohydrate energy, non-smoker | (5) | waist circumference, systolic blood pressure, diastolic blood pressure, sex, age, weight, height, carbohydrate energy, non-smoker | (9) |
| DT | *BPWC_add, *BPWC_mul, *BPWC_dif | (3) | waist circumference, systolic blood pressure, diastolic blood pressure, sex | (4) |
| RF | *WC, *BPWC_dif, CUN-BAE, grain, retinol, kimchi, fat energy | (8) | waist circumference, systolic blood pressure, diastolic blood pressure, sex, age, weight, height, grain, retinol, kimchi, fat energy | (11) |
| XGB | *BPWC_add, *WC, *bWC, *BP, WHR, green vegetables | (6) | waist circumference, systolic blood pressure, diastolic blood pressure, sex, wight, height, hip circumference, green vegetables | (8) |
| TN | *bWC, *BP, leaf tea, lettuce | (4) | waist circumference, systolic blood pressure, diastolic blood pressure, sex, weight, height, leaf tea, lettuce | (8) |
| All | *WC, *BP, *BPWC_add, *BPWC_mul, *BPWC_dif, *bWC, CUN-BAE, WHR, carbohydrate energy, fat energy, non-smoker, grain, retinol, kimchi, green vegetables, leaf tea, lettuce | (17) | waist circumference, systolic blood pressure, diastolic blood pressure, sex, age, wight, hip circumference, height, carbohydrate energy, fat energy, non-smoker, grain, retinol, kimchi, green vegetables, leaf tea, lettuce | (17) |

N: The number of features. The proposed synthetic feature is an asterisk before the feature name. Raw features refer to features of a single piece of information, such as height, weight, and waist circumference, and synthetic features are created using these raw features.

proposed synthetic features. The proposed features are WC, BP, BPWC_add, BPWC_mul, BPWC_dif, and bWC, and the features of previous studies are CUNBAE and WHR. These synthetic features were composed of raw features, as listed in Table 8. From the raw feature point of view, the proposed synthetic features were mainly based on waist circumference, systolic and diastolic blood pressure, and sex, and the features proposed in previous studies consisted of age, sex, weight, height, waist circumference, and hip circumference. The selected lifestyle-related features were carbohydrate energy, fat energy, grain, retinol, kimchi, green vegetables, leaf tea, lettuce, and non-smoker (see Table 9 for the meaning of each feature). Most were related to food or nutrient intake, and only the current smoking status was related to lifestyle.

In all classifiers, our proposed synthetic features were not only selected as important features but also ranked high (Fig 4). Specifically, synthetic features composed of WC, BP, and variations of these two were selected. Existing synthetic features and lifestyle-related features followed our synthetic features: CUN-BAE, WHR, carbohydrate energy, non-smoker, grain, retinol, kimchi, fat energy, green vegetables, leaf tea, and lettuce (see Tables 8 and 9 for the meaning of these features).

Based on the classification model, the total number of features was DT (3) < TN (4) < LR (5) < XGB (6) < RF (8), in ascending order. Based on the number of raw features, the order

**Table 8. Raw features used for synthesis.**

| Synthetic feature | Raw features used for synthesis | N |
|---|---|---|
| WC | waist circumference, sex | 2 |
| BP | systolic blood pressure, diastolic blood pressure | 2 |
| BPWC_add BPWC_mul BPWC_dif | waist circumference, sex, systolic blood pressure, diastolic blood pressure | 4 |
| bWC | waist circumference, sex, weight, height | 4 |
| CUN-BAE | age, sex, weight, height | 4 |
| WHR | waist circumference, hip circumference | 2 |

N: The number of features.

**Table 9. Descriptions of raw features used for synthesis.**

| Feature | Description |
|---|---|
| carbohydrate energy | the percentage of energy obtained from carbohydrates |
| fat energy | the percentage of energy obtained from fat |
| non-smoker | Currently non-smokers (smoking = 0, non-smoking = 1) |
| grain | average daily grain intake |
| retinol | average daily retinol intake |
| kimchi | the average daily intake of kimchi |
| green vegetables | average daily green vegetable intake |
| leaf tea | Average daily green tea intake |
| lettuce | the average daily intake of lettuce |

Kimchi is a traditional Korean dish with pickled cabbage.

differed slightly: DT (4) < XGB = TN (8) < LR (9) < RF (11). DT was the smallest with 4, and RF was the largest with 11. DT used the fewest features in both cases, requiring only four raw features: waist circumference, systolic blood pressure, diastolic blood pressure, and sex. Fig 5 schematically shows the inclusion relationship of the raw features required for predicting each classification model. At its core, there are four features used by the DT, which are characterized by direct involvement in the MetS diagnostic criteria. Along with these four core features, height and weight are key features used in all classifiers, except DT. Based on these key features, each classifier has peripheral features, such as different dietary habits and lifestyles.

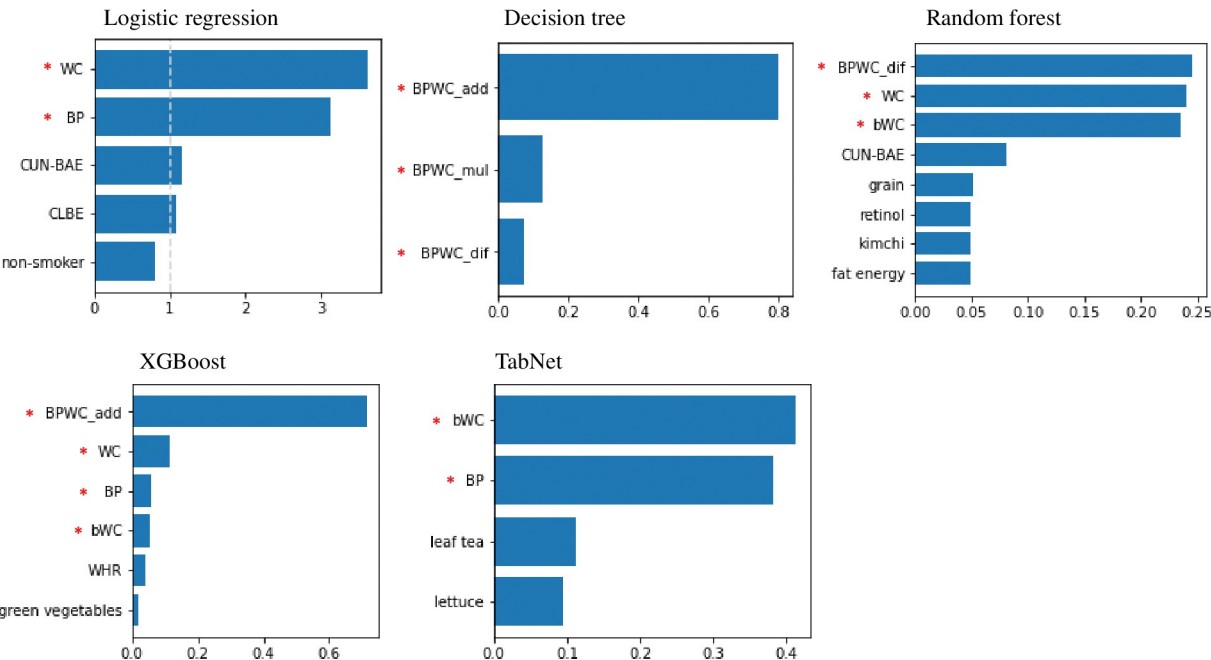

**Fig 4. Optimized classifiers and their important features.** The x-axis is the relative importance of the features, and the y-axis is the name of the features used. The x-axis of LR is the value of applying the regression coefficient to the exponential function. "Our proposed synthetic feature" was asterisked in red before the feature name. (CLBE: The percentage of energy obtained from carbohydrates, non-smoker: Current smoking status, grain: Whole grain intake, retinol: Retinol intake, kimchi: Kimchi (Korean traditional food) intake, fat energy: Percentage of energy from fat, green vegetables, leaf tea: Green tee intake, lettuce: Lettuce intake) See Table 4 for synthetic features.

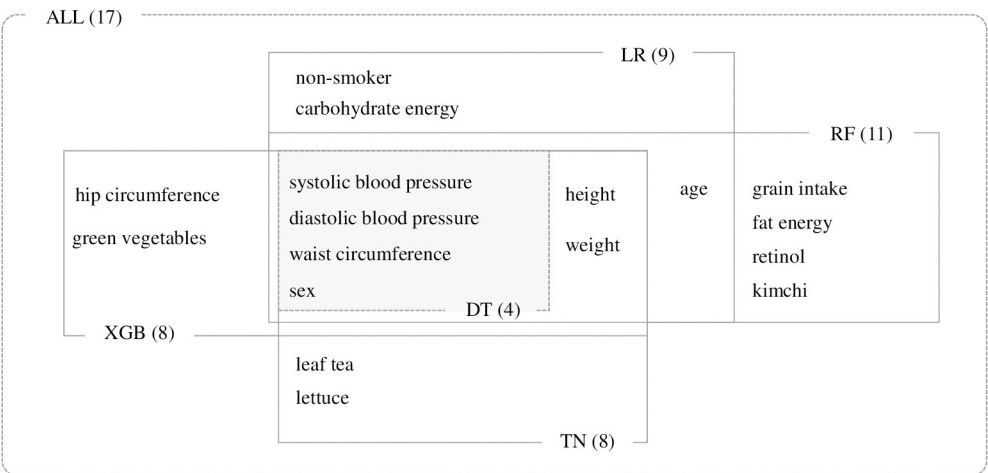

**Fig 5. Feature diagram for each classifier from raw features perspective.** The number of features used by each classifier is indicated in parentheses.

## Candidate models

**Model parameters.** Table 10 lists the results of the parameter tuning for each classifier. These parameters were used to build the final candidate model for each classifier. Table 7 lists the final selected features used by each candidate model.

**Performance comparison.** Table 11 summarizes the performances of the candidate models. Except for LR, the performance of all candidate models improved after parameter tuning. The DT showed the most noticeable performance improvement over the other models, with an AUC of 0.792–0.886. Comparing the models with optimized parameters based on AUC,

**Table 10. Parameters of the candidate models after tuning the parameters.**

| Model | Preset parameters | Optimal parameters |
|---|---|---|
| LR | random_state: 100,<br>penalty: 'none' | solver: 'saga' |
| DT | random_state: 100 | criterion: 'gini',<br>max_depth: 5,<br>max_features: None,<br>min_samples_split: 0.01,<br>splitter: 'best'<br>min_samples_leaf: 200 |
| RF | random_state: 100 | bootstrap: True,<br>max_depth: None,<br>min_samples_leaf: 4,<br>min_samples_split: 10,<br>n_estimators: 2000 |
| XGB | random_state: 100,<br>objective: 'binary:logistic',<br>booster: 'gbtree',<br>n_jobs: -1,<br>seed: None | n_estimators: 500,<br>max_depth: 4,<br>learning_rate: 0.01,<br>subsample: 0.7 |
| TN | seed: 100<br>optimizer_fn:torch.optim.Adam,<br>optimizer_params: dict(lr = 2e-2),<br>scheduler_params:{"step_size":50, "gamma":0.9},<br>scheduler_fn:torch.optim.lr_scheduler.StepLR,<br>mask_type:'entmax' | n_steps: 2,<br>gamma: 1.1,<br>n_independent: 2,<br>n_shared: 3,<br>momentum: 0.03, |

**Table 11. Performance of candidate models.**

| | | LR | DT | RF | XGB | TN |
|---|---|---|---|---|---|---|
| Before Parameter tuning | AUC | 0.887±0.005 | 0.792±0.011 | 0.885±0.005 | 0.884±0.005 | 0.891±0.005 |
| | Recall | 0.818±0.014 | 0.806±0.017 | 0.864±0.014 | 0.857±0.011 | 0.884±0.021 |
| | Specificity | 0.791±0.006 | 0.726±0.008 | 0.758±0.006 | 0.76±0.006 | 0.743±0.023 |
| | BACC | 0.805±0.006 | 0.766±0.008 | 0.811±0.007 | 0.809±0.006 | 0.813±0.007 |
| After Parameter tuning | AUC | 0.887±0.005 | 0.886±0.005 | 0.889±0.004 | **0.893±0.004** | 0.892±0.005 |
| | Recall | 0.818±0.014 | 0.875±0.016 | 0.872±0.013 | **0.89±0.012** | 0.885±0.016 |
| | Specificity | **0.791±0.006** | 0.747±0.014 | 0.756±0.006 | 0.746±0.006 | 0.742±0.014 |
| | BACC | 0.805±0.006 | 0.811±0.006 | 0.814±0.007 | **0.818±0.007** | 0.814±0.006 |

AUC: Area Under Curve; BACC: Balanced Accuracy.

XGB (0.893) performed the best, followed by TN (0.892), RF (0.889), LR (0.887), and DT (0.886). All classifiers achieved a higher recall than specificity. On average, recall and specificity were 0.868 and 0.756, respectively. Recall was highest in XGB (0.89), followed by TN(0.885), DT(0.875), RF(0.872), and LR(0.818). The maximum difference in recall was 0.072, but in tree-based classifiers excluding LR, the difference significantly decreased to 0.018. Conversely, LR showed the highest specificity at 0.791, and the tree classifiers performed close to 0.75.

## Calibration of candidate models

Table 12 and Fig 6 show the results of the evaluation using a test dataset after applying calibrations to the optimal models. First, as shown in Fig 6, when calibration was not applied, the predicted probability in all classification models was overestimated compared to the actual probability. However, there are some differences in each method, but the prediction probability is well corrected by applying the calibration overall. As shown in Table 12, the Brier score

**Table 12. Calibration results for each classification model.**

| Classifier | Metric | Calibration methods | | | |
|---|---|---|---|---|---|
| | | Original | Sigmoid | Istonic | Undersample |
| LR | Brier Score | 0.137 | 0.080 | **0.080** | 0.080 |
| | Spiegelhalter Z-score | 0.029 | 1.566 | **1.398** | 1.321 |
| | Spiegelhalter p-value | 0.977* | 0.117* | **0.162*** | 0.187* |
| DT | Brier Score | 0.137 | 0.079 | 0.079 | **0.078** |
| | Spiegelhalter Z-score | 5.851 | 0.875 | 2.143 | **0.352** |
| | Spiegelhalter p-value | 0.00 | 0.382* | 0.032 | **0.725*** |
| RF | Brier Score | 0.137 | **0.079** | 0.079 | 0.079 |
| | Spiegelhalter Z-score | 4.456 | **1.126** | 1.968 | -0.006 |
| | Spiegelhalter p-value | 0.00 | **0.260*** | 0.049 | 0.995* |
| XGB | Brier Score | 0.136 | 0.078 | **0.077** | 0.077 |
| | Spiegelhalter Z-score | 4.721 | -0.180 | **1.113** | -1.035 |
| | Spiegelhalter p-value | 0.00 | 0.857* | **0.266*** | 0.301* |
| TN | Brier Score | 0.132 | **0.078** | 0.078 | 0.078 |
| | Spiegelhalter Z-score | 7.280 | **0.721** | 1.931 | 2.257 |
| | Spiegelhalter p-value | 0.00 | **0.471*** | 0.054* | 0.024 |

The calibration was considered significant if the p-value was greater than 0.05; * indicates significance. For each classifier, the final selected calibration result is bolded.
Original: Non-calibration, Sigmoid: Platt scaling, istonic: Isotonic regression, Undersample: Pozzolo's method.

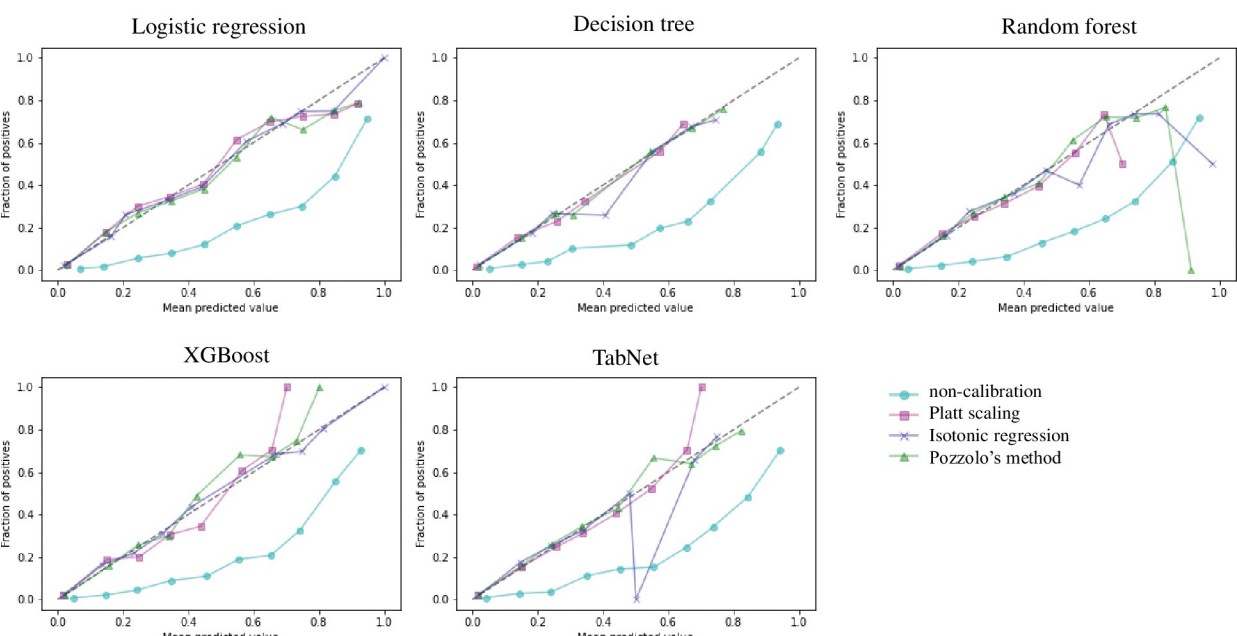

**Fig 6. Calibration plot for each classification model.** The x-axis is the mean predicted value, and the y-axis is the fraction of positives. The more it matches the diagonal line of the plot, the better the calibration. If the plot is drawn diagonally below, the predicted result is overestimated than the actual result.

of the calibrated model was lower, and there was no significant difference between the calibration methods. However, there was a difference in the significance of the results between calibration methods when evaluated based on Spiegelhalter's statistics. DT and RF were not significant in the istonic regression method, and TN was not significant in Pozzolo's method. Considering the calibration plot, Brier score, and Spiegelhalter's statistics, we determined that it was appropriate to calibrate RF and TN with Platt scaling, LR and XGB with istonic regression, and DT with Pozzolo's method.

## Comparison of candidate models

The characteristics of the calibrated models based on the analysis thus far are summarized in Table 13. The characteristics of the model were compared in terms of four aspects:

**Table 13. Final evaluation results of candidate models.**

|  | LR | DT | RF | XGB | TN |
|---|---|---|---|---|---|
| AUC | 0.89 | 0.889 | 0.893 | 0.896 | 0.893 |
| Recall | 0.838 | 0.855 | 0.878 | 0.886 | 0.866 |
| Specificity | 0.792 | 0.773 | 0.761 | 0.756 | 0.765 |
| BACC | 0.815 | 0.814 | 0.819 | 0.821 | 0.815 |
| Used raw feature (N) | 9 | 4 | 11 | 8 | 8 |
| Preprocessing required | Yes | No | No | No | No |
| Calibration method | Istonic | Undersample | Sigmoid | Istonic | Sigmoid |
| -Brier score | 0.080 | 0.078 | 0.079 | 0.077 | 0.078 |
| - Spiegelhalter Z-score | 1.398 | 0.352 | 1.126 | 1.113 | 0.721 |
| - Spiegelhalter p-value | 0.162 | 0.725 | 0.260 | 0.266 | 0.471 |
| Interpretability | Easy | Easy | Hard | Hard | Feasible |

discrimination, calibration, ease of use of features, and interpretability. All the models were evaluated using the same test dataset.

From the perspective of discrimination, XGB exhibited the highest performance at 0.896. RF and TN(0.893), LR(0.89), and DT(0.889) followed. However, the gap between the best and poor performances was 0.007, and the difference in discrimination between the models was not noticeable. Recall, specificity, and balancing accuracy showed similar patterns to AUC. In terms of calibration, it also showed significant performance without notable differences between the models.

When comparing the number of raw features required for prediction, the DT was the smallest with four, followed by XGB, TN, LR, and RF. The top three performance models, XGB, TN, and XGB, used more than eight raw features. The DT only required raw features of less than half of the other prediction models. Furthermore, unlike LR, DTs have the convenience of not requiring a preprocessing process, such as scale, when using features.

Each model was calibrated using different calibration methods, and the results were significant when evaluated using the Brier score, Spiegelhalter z-score, or p-value. Thus, in all models, the predictive probability can be interpreted as the actual probability, that is, the risk of developing MetS. In addition to the interpretability of predictive probabilities, LR, DTs, and TN are characterized by easy interpretation of the model itself. On the other hand, RF and XGB have poor interpretation of predictive results in an ensemble of numerous trees.

## Decision of final MetS predictive model

The five classification algorithms produced prediction models with similar performances. In this case, the simpler the model, the better. Therefore, the number of features used was the criterion for the model selection. The decision tree used the fewest raw features (systolic and diastolic blood pressures, waist circumference, and sex) compared to the other models, and these features were also easy to collect. Furthermore, DTs have several advantages: first, they do not need assumptions about data such as LR; second, they can be used directly as predictors without preprocessing such as scaling; third, they are easy to interpret as the model itself structurally internalizes the decision process. RF, XGB, and TN are tree-based models that can be used without preprocessing and have nonparametric model properties, but RF and XGB are difficult to interpret as ensemble models. On the other hand, TN can interpret the prediction results by instance, but it is not as intuitive as a DT. Therefore, we determined that the DT is a practical model with many advantages over performance when comprehensively considering discrimination, calibration, ease of use of features, and interpretability.

Fig 7 shows an example of how the final selected decision-making model works. When the user provides information on systolic blood pressure, diastolic blood pressure, waist circumference, and sex as input values, these four pieces of information become raw features and are converted into synthetic features called BP and WC. BP and WC were synthesized once more to produce three synthetic features, BPWC_add, BPWC_mul, and BPWC_dif, which were used as the final input values for the prediction model. The model then outputs the predictive results of whether this user has MetS, what is the probability, and how many times the probability of developing the disease compared to the average. For example, if a woman had a systolic blood pressure of 140 *mmHg*, diastolic blood pressure of 90 *mmHg*, and waist circumference of 89 *cm*, these measurements were first converted to 0.84 (BP) and 0.66 (WC). After that, the conversion values are once again converted to 1.50 (BPWC_add), 0.55 (BPWC_mul), and 0.18 (BPWC_dif) and input into the model. Based on this input value, the model diagnosed MetS with a probability of 0.31 (risk) and provided information that the risk probability was 2.25 times more likely to develop than the average.

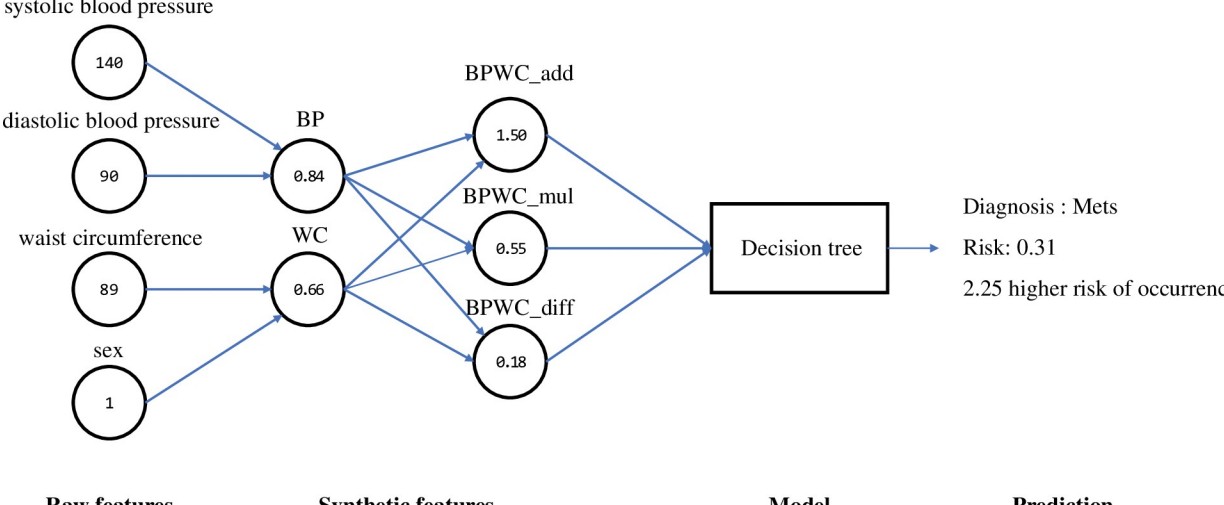

**Fig 7. Example of the execution process of the final model from raw feature to prediction.** The values in the circle are the actual values for an instance.

## Decision tree and metabolic syndrome risk map

We devised a "MetS risk map" with WC and BP as axes by interpreting the structured results of the decision-making process. The decision tree outputs the result of structuring the decision process in the form of the plot (A) or text (B), as shown in Fig 8. We decomposed the classification rules for each node, as shown in Fig 8B, and expressed them on a plane with WC and BP as axes. This was possible because the DT model used only three features represented by the relationship between WC and BP: BPWC_mul = BP * WC, BPWC_add = BP + WC, and BPWC_dif = BP-WC. DTs divide the space using vertical or horizontal lines; however, we were able to divide the space by diagonal and curve using the relationship of these features.

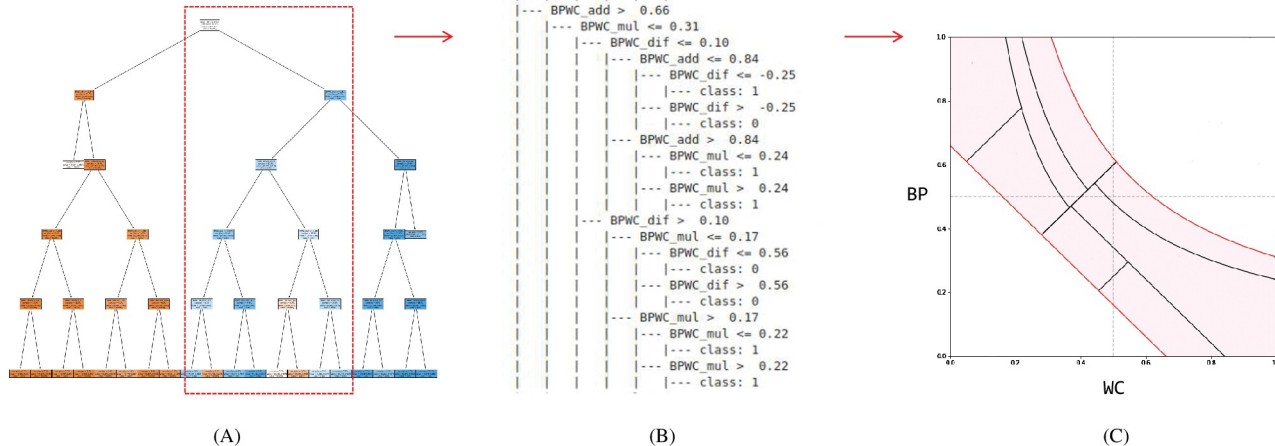

**Fig 8. Creating a MetS risk map from the decision tree.** (A) is the DT of this study and has a depth of 5. Each node is a classification rule for datasets, with blue representing MetS and orange representing non-MetS. The higher the probability, the darker is the color. (B) is a textual representation of the decision rule in (A), where only the red box portion is taken. Classification rules are expressed in the form of inequality; class 1 means MetS, and class 0 means non-MetS. (C) represents the inequality expressed in (B) in a plane. For example, at (C), the red diagonal at the bottom is the line for BPWC_add = 0.66. Because BPWC_add = BP + WC, the rule can be drawn on a plane with WC and BP as axes.

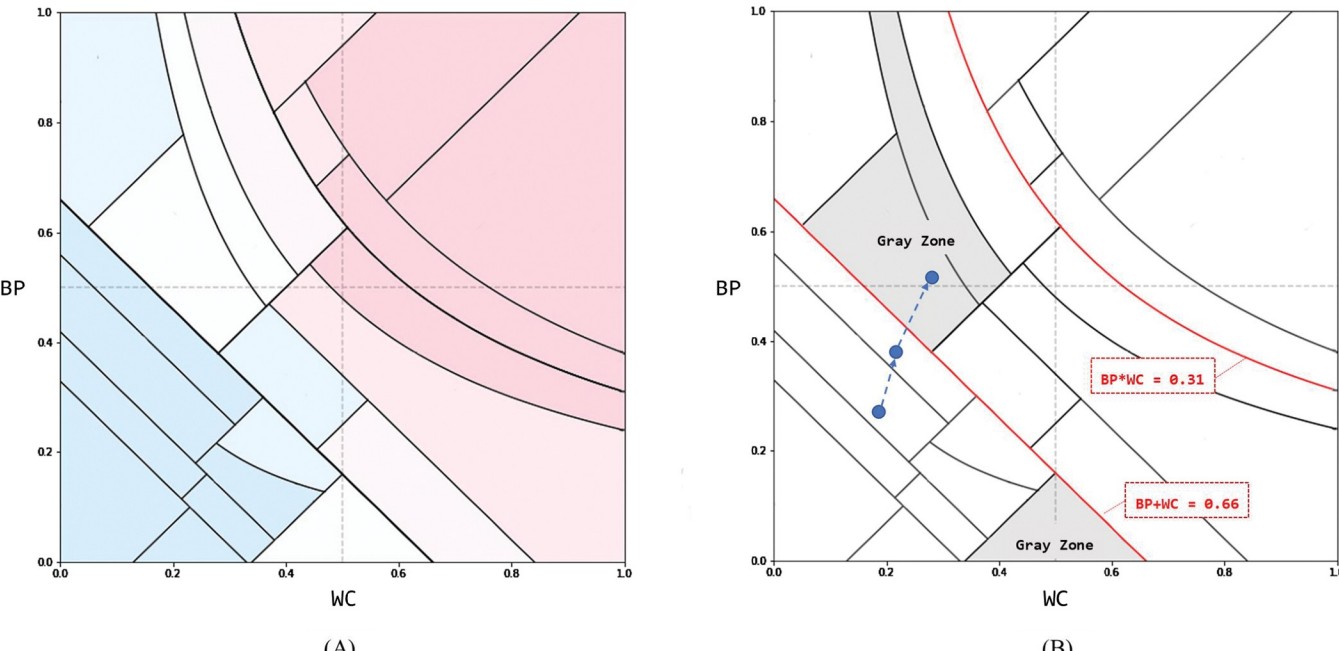

**Fig 9. MetS risk map.** (A) is the final form of the MetS risk map. We split the region to correspond to the leaf node of the DT on a plane with the proposed synthetic feature WC and BP as two axes. Within each region, there is a record of how much the incidence is higher than average. The greater the value, the higher is the risk of developing the disease. Regions with values of 1 or more are classified as MetS areas and marked in red. In (B), important lines and regions are emphasized in preventing and managing MetS management.

Each divided region of the MetS risk map corresponded one-to-one to the terminal node of the DT. The MetS risk map was completed by matching the risk of MetS using the calibrated probability for each region (Fig 9). Risk is the calibrated probability divided by an adjusted threshold. The threshold was adjusted using Pozzolo's method [28] and was found to be 0.137, similar to the prevalence of MetS in the population. Therefore, the risk can be interpreted as the number of times the probability of incidence is higher than the prevalence of MetS in the entire population.

The MetS risk maps were divided into three zones. These zones were formed by two lines, as shown in Fig 9B. The first zone is the lower part of the area divided by BP+WC =0.66 and is a safety zone for MetS. Most regions were classified as non-MetS, and the risk of development was much lower than 1. The second zone is the upper part of the area divided by BP×WC =0.31 and is a risk zone for MetS. All regions were classified as having MetS, and the risk of development was > 2. The third zone is the area between the two lines and is a warning zone. This zone gradually progresses to MetS and is the most important zone for prevention. In more detail, the important region for prevention can be narrowed down to the region indicated by the gray zone in Fig 9B. The gray zone is the region where the risk increases rapidly compared to the adjacent non-MetS regions. At the same time, MetS and non-MetS existed at similar rates in this area. Combining these two facts, we can infer that the gray area is the path to active conversion to MetS.

## Summary of results

In the results section, we describe various aspects of the developed MetS prediction models. The summary is as follows:

① Our proposed synthetic features were effective in enhancing the classification performance. Specifically, synthetic features based on BP and WC were evaluated as being the most important among all classifiers. The synthetic feature uses only waist circumference, systolic and diastolic blood pressures, and sex as the base features, which includes all classifiers in common.

② In the analysis of the predicted probability of the models, we found a tendency to overestimate MetS in all classifiers and calibrated it to reduce the estimation error. Therefore, the probability predicted by the calibrated model was indicative of the risk of developing MetS.

③ We selected the DT model as the final predictive model for MetS. It used the fewest features for prediction but derived an almost similar performance to the other models. Four raw features, namely waist circumference, systolic and diastolic blood pressures, and sex, were used, which have the advantage of being easily measured in daily life. The decision tree model is simple and has transparent properties that can be used to understand the decision structure.

④ We devised a MetS risk map by reconstructing the decision structure of the final model as a two-dimensional plane and mapping the risk probability to each region.

## Discussion

We developed a predictive model for MetS that utilizes only noninvasive information, making it practical for use in real-world scenarios. While fasting blood sugar, triglycerides, and HDL cholesterol are important factors in diagnosing MetS, we deliberately excluded features that require blood testing when developing our predictive model, to ensure its preventive usability.

The proposed model has three major advantages for the preventive management of MetS. The first advantage is that the features required for prediction are just four easily measurable features: waist circumference, systolic and diastolic blood pressure, and sex. Second, the predictive model provides the degree of risk along with the diagnosis of MetS, enabling individuals to cope effectively with preventive management. Third, prediction results can be easily understood by individuals, and prediction models can be provided as visual tools reconstructed in a simple map form. These three advantages are also consistent with the technology acceptance model (TAM), which is a theory on the properties of information technology to be well received in society. According to TAM, the higher the perceived usefulness and perceived ease of use of technology, the higher the acceptability of the technology [31]. Perceived usefulness is related to usefulness and productivity for the task, and perceived ease of use has been embodied, such as clear, understandable, and low mental effort [32]. Perceived usefulness is also affected by perceived ease of use; that is, it is recognized as more useful when the user is easy to use [33].

Using five classification algorithms, we identified 17 noninvasive raw features useful for predicting MetS (Fig 5). At the center, systolic and diastolic blood pressures, waist circumference, and sex were directly related to MetS diagnostic criteria. We synthesized four key anthropometric features to create BP and WC features. These novel features, including various variants using BP and WC, were of higher importance in predicting MetS than synthetic features, such as CUN-BAE, BRI, and BMI, as proposed in previous studies. This result was presumed to be due to the inherent properties of BP and WC. The synthetic features reflect a certain section that is more important around the diagnostic criteria, as shown in Fig 10. The two axes, WC and BP, of the MetS risk map can be interpreted similarly to the distance from the diagnostic criteria of abnormal factors in MetS. To be exact, BP and WC are values that

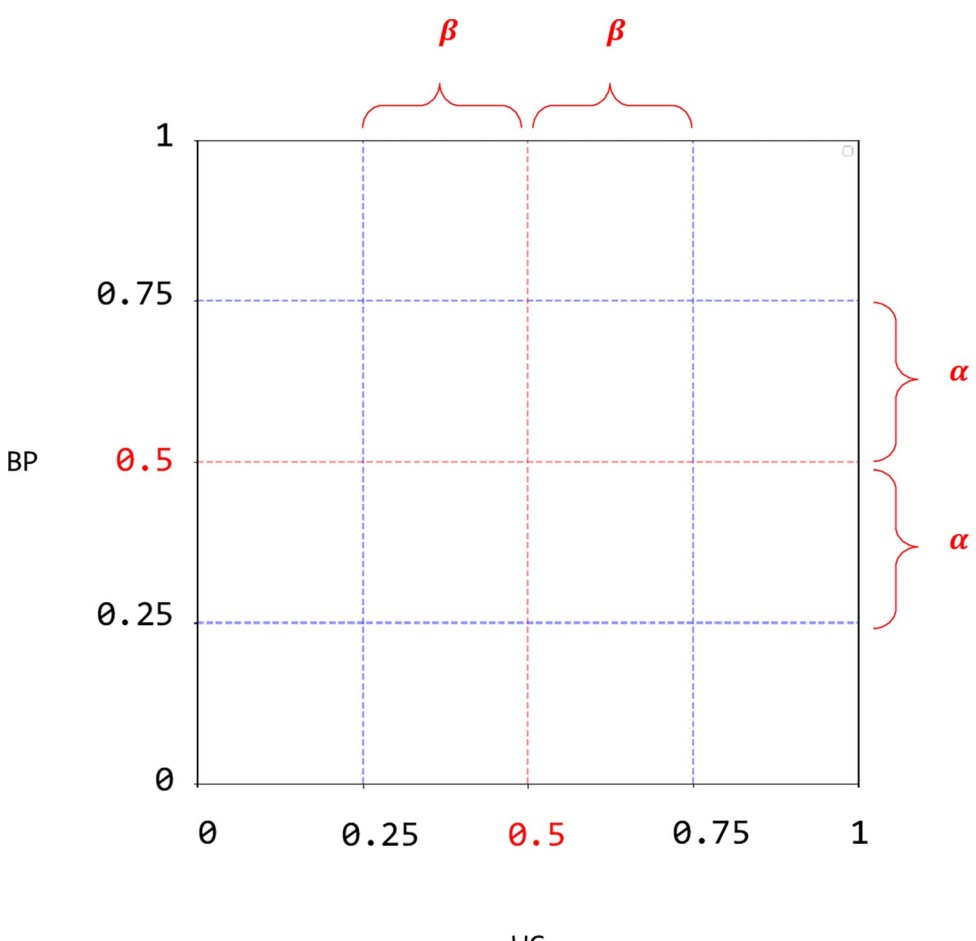

**Fig 10. Coordinate space of composite feature WC and BP.** The 0.5 point of the two axes is equal to the diagnostic criterion. WC is a feature synthesized by waist circumference and gender. BP is a feature synthesized by systolic and diastolic blood pressure. $\alpha$ = (diagnostic criteria for systolic blood pressure–diagnostic criteria for diastolic blood pressure)*0.1, $\beta$ = *Diagnostic criteria for waist circumference* *0.1.

consider about 10% of the diagnostic criteria more important and have nonlinearity similar to sigmoid, expressed as a value between 0 and 1. BP and WC had an actual diagnostic criterion of 0.5. Therefore, if this value approaches 0.5, it is close to the diagnostic criterion, and it exceeds the diagnostic criterion if it exceeds 0.5. Furthermore, the 0.25–0.75 interval, which is half the value between 0 and 1, corresponds to 10% of the diagnostic value before and after the diagnostic criterion. That is, BP is the weighted position of blood pressure with respect to the diagnosis criterion, and WC is the weighted position of waist circumference with respect to the diagnosis criterion. This property fits well with the perspective of preventing chronic diseases. It is more effective in the prevention of looking at a certain section with greater risk than looking at all steps with the same importance.

Lifestyle-related features were not evaluated as important as anthropometric features. This result was also reported in previous studies [4–13]. However, given that many studies have reported an association between lifestyle and MetS, we speculate that the way lifestyle-related information is collected was not sufficient to reveal its characteristics. In fact, Tabares et al. [34] recently reported that increasing physical activity levels and lowering BMI by at least 2% reduced the risk of developing MetS by 3.8% but added that increasing physical activity

without weight loss had little effect on prediction. This finding disproves that lifestyle influences are observable when accompanied by meaningful physical changes. Therefore, it is necessary to examine whether the lifestyle data used in this study contain sufficient information accompanying physical changes. In the case of the dietary data used in this study, the frequency of food intake was collected through questionnaires, which was a form of responding to the monthly/weekly/day unit while recalling the "average frequency of intake over the past year" [35]. If it is collected several times in cycles shorter than a year, we expect it to be different from the current results. In fact, a study using three follow-up datasets reported that the performance was improved by using cumulative survey data [6]. In addition, precise and dense lifestyle data are expected to accumulate as healthcare technologies, such as smartwatches, smart bands, and diet management apps have become popular. Therefore, future studies are needed to identify important lifestyle features for MetS prediction based on these data.

The DT finally achieved an AUC of 0.889, recall of 0.855, and specificity of 0.773. Compared to previous studies, its performance is difficult to compare directly because of differences in race, population size, and prevalence of MetS, but it is similar in terms of AUC (see Table 1). However, our model was characterized by higher recall than specificity. Individuals without MetS are more likely to be classified as having MetS compared to previous research models, but from the standpoint of preventive management, it is appropriate to conservatively diagnose suspected patients and induce additional checkups. In addition, when comparing studies on the MetS prediction model of Koreans over the past decade [36], to the best of our knowledge, this study is the first based on noninvasive information from large-scale Koreans.

Of the studies listed in Table 14, two studies ([9,13]) identified gender-specific differences in model performance when considering prevalence-robust metrics. Specifically, male models outperformed female models in both studies, with a male balanced accuracy of 0.807 and female of 0.646 in [13] and a male recall of 0.594 and female of 0.409 in [9]. While key anthropometric features were consistent across genders, variations were observed in food-related features. Therefore, we developed the final decision tree (DT) model separately for men and women and compared their performance (Table 14): the balanced accuracy was 0.872 for men and 0.890 for women, and the recall was 0.843 for men and 0.850 for women. Both gender models shared common features such as BPWC_add and BPWC_mul (Fig 11), while the female model included additional anthropometric and dairy-related food features. However, when tested on the same dataset, the integrated model performed better for men based on AUC, while no significant difference was observed for women. Consequently, the study concludes that the individual models' impact on feature selection and performance was insignificant.

From a positive predictive value (PPV) perspective, it is possible to divide the risk map into three distinct areas: green, yellow, and red (Fig 12A). The green zone refers to a section that is not associated with MetS, where the risk of developing metabolic syndrome is 0 to 1 times or lower than the average prevalence of 13.7%. The yellow zone refers to a section in which the

**Table 14. Performance of integrated and sex-specific individual models using the same test set.**

| Case | | AUC | Recall | Specificity | BACC | Features |
|---|---|---|---|---|---|---|
| Integrated Model | All | 0.889 | 0.855 | 0.773 | 0.814 | BPWC_add, BPWC_mul, BPWC_dif |
| | Male | 0.878 | 0.895 | 0.690 | 0.792 | |
| | Female | 0.890 | 0.826 | 0.806 | 0.816 | |
| Individual model | Male | 0.872 | 0.843 | 0.736 | 0.790 | BPWC_add, BPWC_mul, WC |
| | Female | 0.890 | 0.850 | 0.780 | 0.815 | BPWC_add, BPWC_mul, bWC, CUNBAE, BFP, dairy products |

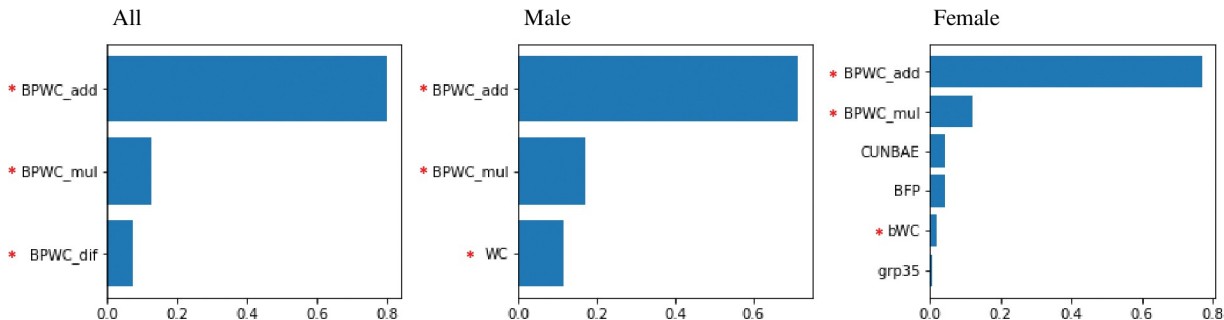

**Fig 11. Features and feature importance of integrated and sex-specific individual models.** "grp35" is a synthetic feature related to dairy products.

risk of developing MetS is 1 to 2.5 times higher. This transitional phase represents a mixed area of cases, including both non-MetS and MetS. Despite its ambiguity due to a low PPV of 0.248, it is imperative to prioritize the prevention and management of MetS in this state, given the increased risk probability compared to the average. The red zone refers to a section in which the risk of developing MetS is 4 to 5.7 times higher. With a PPV of 0.673, this area is deemed as having a high likelihood of actual metabolic syndrome cases, necessitating blood tests for accurate diagnosis.

Table 15 summarizes the positive predictive values (PPV) for each gender in different zones. The Yellow Zone indicates relatively low PPV for both males (0.258) and females (0.241), implying that only one out of four predicted individuals are likely to have metabolic syndrome. In contrast, the Red Zone exhibits higher PPV for males (0.725) and females

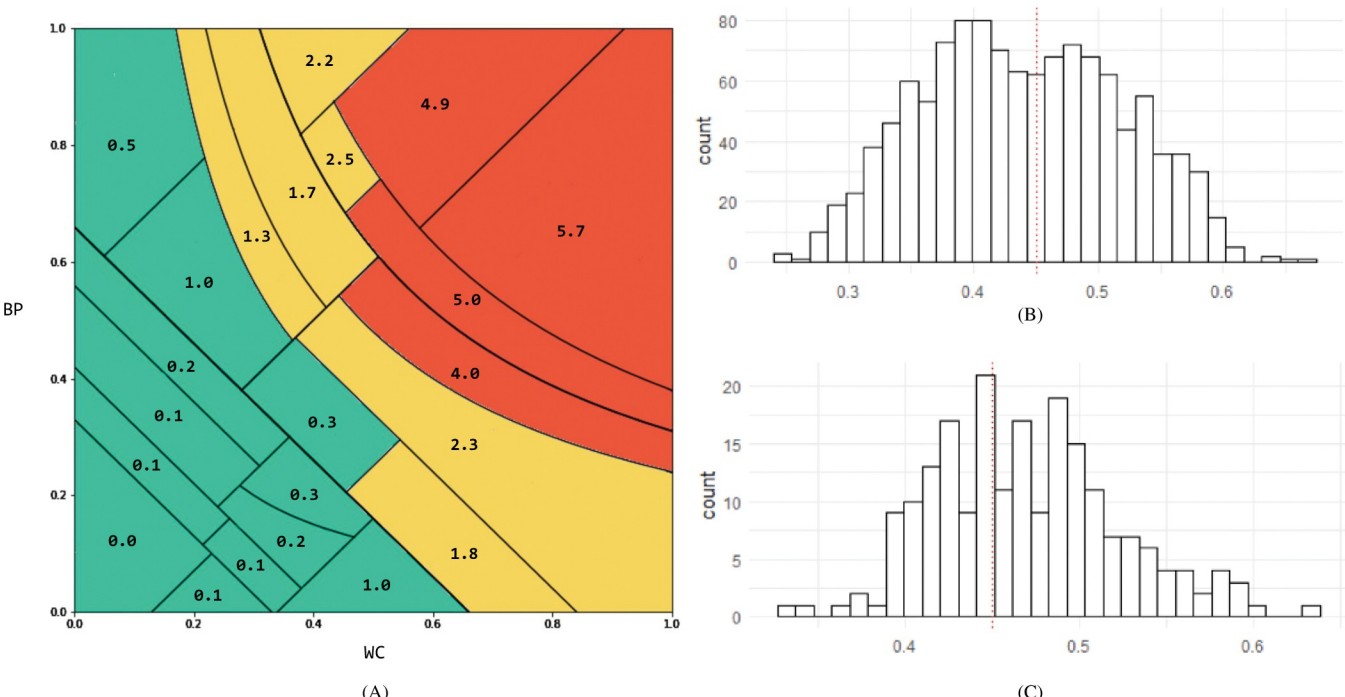

**Fig 12. Risk map from PPV perspective and the risk index distribution in FP cases.** (A) Three zones of risk map from the perspective of PPV perspective: Green, yellow, and red zone. (B) Risk index distribution of FP in yellow zone. (C) Risk index distribution of FP in red zone. 0.45 is the point where MetS management is required preemptively (indicated by a red dotted line).

**Table 15. Misclassification by zone.**

| Zone | Group | FN | FP | TN | TP | PPV | NPV | NT-> TP | Adjusted PPV |
|---|---|---|---|---|---|---|---|---|---|
| Green+Yellow+Red | ALL | 137 | 1382 | 4,707 | 811 | 0.370 | 0.972 | 675 | 0.678 |
| | Male | 42 | 534 | 1,187 | 359 | 0.402 | 0.966 | 273 | 0.708 |
| | Female | 95 | 848 | 3,520 | 452 | 0.348 | 0.974 | 302 | 0.580 |
| Green | ALL | 137 | 0 | 4,707 | 0 | - | 0.972 | - | - |
| | Male | 42 | 0 | 1,187 | 0 | - | 0.966 | - | - |
| | Female | 95 | 0 | 3,520 | 0 | - | 0.974 | - | - |
| Yellow | ALL | 0 | 1176 | 0 | 387 | 0.248 | - | 552 | 0.582 |
| | Male | 0 | 458 | 0 | 159 | 0.258 | - | 229 | 0.629 |
| | Female | 0 | 718 | 0 | 228 | 0.241 | - | 223 | 0.551 |
| Red | ALL | 0 | 206 | 0 | 424 | 0.673 | - | 123 | 0.868 |
| | Male | 0 | 76 | 0 | 200 | 0.725 | - | 44 | 0.884 |
| | Female | 0 | 130 | 0 | 224 | 0.633 | - | 79 | 0.856 |

FN: False Negative, FP: False Positive, TN: True Negative, TP: True Positive, NPV: Negative Predictive Value, PPV: Positive Predictive Value. "NP->TP" is the number of cases that can be reclassified from NP to TP when determining whether or not to be prevented from MetS based on the risk index 0.45. Adjusted PPV is the value calculated by reflecting "NP->TP."

(0.633) compared to the Yellow Zone, with males having a higher PPV. These findings suggest that 6 to 7 out of 10 predicted individuals are highly likely to have metabolic syndrome. However, the use of binary classification diagnostic criteria to calculate PPV is inadequate for the prevention of metabolic syndrome, which is a chronic disease that develops progressively over time. Our focus, therefore, should be on providing individuals with opportunities to manage the condition before it progresses to a more severe stage. Thus, in this study, we analyzed FP from the perspective of severity, rather than solely on the presence of metabolic syndrome. To achieve this, we employed the risk index proposed in our previous work [18] and obtained a risk distribution for the FP cases. The resulting risk distribution is presented in Fig 12B and 12C. Our risk index employs a diagnostic threshold of 0.547 for identifying MetS [18]. However, MetS may also occur at values as low as 0.45, leading us to classify individuals in the MetS risk group as having a score of 0.45. Using this criterion, we identified 552 of the 1176 FP cases in the Yellow Zone as true positives, resulting in an adjusted PPV of 0.582, up from 0.248. Similarly, in the Red Zone, 123 FP cases were reclassified as true positives, resulting in an adjusted PPV of 0.686, up from 0.673. Nevertheless, to enhance PPV significantly, additional research is imperative to enhance the performance by configuring individual models for each zone or region exhibiting multiple misclassifications.

Although there is a limit to the low PPV, our final model can effectively help the decision-making process in preventing and managing MetS by providing development risks as well as good discrimination and recall. Previous studies have focused on the diagnosis itself, and the evaluation of the prediction probability has been overlooked. Some studies conducted a calibration for the predicted probability but did not theoretically present a threshold to distinguish the presence or absence of MetS after correction. However, this study further expands the interpretation of the results by calibrating the overestimated prediction probability using the method proposed by Pozzolo [28]. In addition to being able to interpret the predicted probability as a MetS risk, it was also possible to present how serious the state is based on theoretically clear thresholds.

In addition to semantic interpretations of predictive probabilities, such as risk, we devised a way to explain the rationale for predictive outcomes. MetS risk maps are designed to provide

synergy by gathering them in one place with the structural interpretability of the DT, the meaning of the proposed synthetic features (BP and WC) themselves, and prediction probabilities that can be interpreted as risks. MetS risk maps provide a clear guide to healthcare by representing two boundaries where health conditions change significantly. Based on these boundaries, each individual can have a clear perception of whether they are in a safe zone, warning zone, or risk zone for MetS. In addition, the MetS risk map forms a gray zone (Fig 9B) where the conversion of MetS begins in earnest so that the subjects in the zone can take or receive more active management measures. Clinicians can recommend and effectively explain appropriate tests and treatments to patients by referring to the predicted risk for each region of the MetS risk map. Finally, the MetS risk map can also be used as a visual tool to monitor MetS, such as the blue dots in Fig 9B, when measured periodically because one prediction result is represented in plane coordinates.

The results of this study are limited to the characteristics of the population such as race, age, and prevalence. Therefore, our results may not be generalizable to other populations. We used a dataset of middle-aged Koreans in their 40s and 60s and found that the prevalence of MetS was 13.6%. We also excluded subjects undergoing blood pressure and cholesterol-related treatment and those taking blood pressure and cholesterol-related drugs. Moreover, it should be noted that the analysis of this study is based on datasets collected from 2004–2013, so there is a time difference of more than 10 years. However, we expect future studies to allow us to compare and evaluate the performance of other populations because model development follows a procedure to maintain representation within a given population.

## Supporting information

**S1 Fig. MetS risk map with raw values.** The WC axis, representing waist circumference, displays distinct values based on gender (M for males and F for females). The BP axis transforms systolic (S) and diastolic (D) blood pressure values into equivalent values between 0 and 1, respectively, and ultimately selects the larger of the two. The raw values displayed in S, D, M, and F are rounded to one decimal place, which corresponds to the values transformed to a range between 0 and 1. As an example, for females with a waist circumference of 83 cm, systolic blood pressure of 131 mmHg, and diastolic blood pressure of 88 mmHg, the corresponding values for waist circumference, systolic blood pressure, and diastolic blood pressure are 0.4, 0.6, and 0.7, respectively, and the final blood pressure value is 0.7.
(DOCX)

**S2 Fig. An example of a region being divided into decision boundaries.**
(DOCX)

**S1 Table. Structure of the final decision tree.** The ruleset on the left of the decision tree is the final model that was used to construct the risk map in this study. The ruleset on the right is a reconstruction of the left ruleset (BPWC_add, BPWC_dif, BPWC_mul) into BP and WC. Each rule serves as a decision boundary that divides a two-dimensional plane. For instance, the first two rule sets, BP + WC < = 0.66 and BP—WC < = -0.34, form two decision boundaries (red and blue) in S2 Fig, resulting in a gray area.
(DOCX)

## Author Contributions

**Conceptualization:** Hyunseok Shin, Simon Shim, Sejong Oh.

**Data curation:** Hyunseok Shin.

**Formal analysis:** Hyunseok Shin.

**Funding acquisition:** Sejong Oh.

**Investigation:** Hyunseok Shin.

**Methodology:** Hyunseok Shin, Sejong Oh.

**Project administration:** Sejong Oh.

**Resources:** Hyunseok Shin.

**Software:** Hyunseok Shin.

**Supervision:** Simon Shim, Sejong Oh.

**Validation:** Hyunseok Shin.

**Visualization:** Hyunseok Shin, Sejong Oh.

**Writing – original draft:** Hyunseok Shin.

**Writing – review & editing:** Simon Shim, Sejong Oh.

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
