## [Decision Letter · Decision Letter 0]

5 Feb 2023

PONE-D-23-00127Machine leaning-based predictive model for prevention of metabolic syndromePLOS ONE

Dear Dr. Shin,

Thank you for submitting your manuscript to PLOS ONE. After careful consideration, we feel that it has merit but does not fully meet PLOS ONE’s publication criteria as it currently stands. Therefore, we invite you to submit a revised version of the manuscript that addresses the points raised during the review process.

The following should be considered when resubmitting the revised manuscript.<ul> <li> 

Provide a justification and the criteria for selecting the five algorithms.

 <li> 

Use additional feature selection methods e.g. filter-based ones to improve the statistical significance of the features identified in this study.

 <li> 

Since Sex feature was identified as a core feature and used by all classifiers, this indicates that males and females may have different risk factors for MetS. Hence it is therefore possible, if considered separately, to identify specific features for males and females that can increase the accuracy of the ML models. Consider stratifying the data by Sex and run the models separately.

 <li> 

The positive predictive value (PPV) of the DT model, based on the confusion matrix in Figure 10, is 37%. This low value demonstrates the model's inability to predict the presence of MetS in a patient and possibly put the patient at risk associated with unnecessary heavy treatment and strict dieting. Although the authors presented a warning zone using a risk map, the boundaries of the zone are wide and can put patients at risk. If the data was stratified by Sex and the PPV for males and females models be compared to understand whether the proportions of misclassified cases are among males and females.

 <li> 

Another study that was conducted on Korean participants (https://doi.org/10.3390/app10217741). reported the association of triglycerides with MetS, a feature not detected by any of the ML algorithms employed in the manuscript. This study may also invalidate the authors' claim regarding the comparability of the manuscript's findings with other published studies on the fact of race. The author should elaborate and discuss the findings with the results presented in the link above.

 <li> 

Samples were collected between 2004 and 2013. However, the analysis was conducted in 2022. Authors should clearly indicate why the samples from 2014-2022 was excluded from the analysis as it was highlighted that there was an increase in cases in 2018.

 <li> 

The code should be uploaded on github or in another public space for reproducibility purposes.

 <li> 

The language and style corrections mentioned by the reviewers should be corrected, including the misspelling of “machine learning” in the title

We look forward to receiving your revised manuscript.

Kind regards,

Shakuntala Baichoo, Ph.D

Academic Editor

PLOS ONE

Journal Requirements:

"The data used in this study were obtained from the Korean Genome and Epidemiology Study (KoGES; 4851-302), National Institute of Health, Korea Disease Control and Prevention Agency, Republic of Korea."

"This study was supported by the Ministry of Science, ICT (MSIT), Korea, under the High-Potential Individuals Global Training Program (2021-0-01531) and the R&D program of Development of AI ophthalmologic diagnosis and smart treatment platform based on big data(2018–0-00242) supervised by the Institute for Information & Communications Technology Planning & Evaluation (IITP). The funders had no role in study design, data collection and analysis, decision to publish, or preparation of the manuscript."

Additional Editor Comments:

Cross validation using the chosen algorithms and additional metrics like LR+ and LR- would provide better insights about the different ML algorithms. The figures should be improved for clarity.

Best wishes

Reviewers' comments:

Reviewer's Responses to Questions

**Comments to the Author**

1. Is the manuscript technically sound, and do the data support the conclusions?

Reviewer #1: Yes

Reviewer #2: Partly

2. Has the statistical analysis been performed appropriately and rigorously?

Reviewer #1: Yes

Reviewer #2: Yes

3. Have the authors made all data underlying the findings in their manuscript fully available?

Reviewer #1: No

Reviewer #2: No

4. Is the manuscript presented in an intelligible fashion and written in standard English?

Reviewer #1: Yes

Reviewer #2: No

5. Review Comments to the Author

Reviewer #1: The manuscript by Hyunseok et al studied Metabolic Syndrome (MetS) in Korean men and women through the analysis of a data set with features extracted from health check records. The features were grouped into three categories; anthropometric, survey-based and synthesized. The central approach was to apply machine learning (ML), wrapper-based feature selection, on the dataset and propose an ML model on the important features with the best predictive power by AUC, recall and specificity. In general, all ML algorithms applied in the study reported similar predictive power. However, the decision tree (DT) algorithm reported the fewest features, making the model less complex and easier to interpret. The number of features used by the model were 3 and all were synthesized based on the following features: waist circumference, systolic blood pressure, diastolic blood pressure, and sex. These 4 raw features were common among the other model predictors.

Major concerns:

1. Based on the large sample size (n=70,370) it can be expected that the different ML algorithms will exhibit comparative performance metrics. In addition, we can already see in Table 1 that DT in several studies used the fewest features for the prediction of MetS. It is unclear from the manuscript the reason for using five algorithms and what the criteria for selecting the algorithms are.

2. The authors used Feature Selection Wrapper on the five ML algorithms to select the best combination of features based on model performance neglecting the relationship between the selected features and the response variable. Thus, the use of filter-based methods for feature selection will add a statistical significance to the features identified in this study.

3. Sex feature was among the core features used by all classifiers, hence males and females may have different risk factors for MetS. It is therefore possible, if considered separately, to identify specific features for males and females that can increase the accuracy of the ML models.

4. The positive predictive value (PPV) of the DT model, based on the confusion matrix in Figure 10, is 37%. This low value demonstrates the model's inability to predict the presence of MetS in a patient and possibly put the patient at risk associated with unnecessary heavy treatment and strict dieting. Although the authors presented a warning zone using a risk map, the boundaries of the zone are wide and can put patients at risk. Related to the previous comment, wondering what the proportions of misclassified cases are among males and females?

5. This is another study that was conducted on Korean participants (https://doi.org/10.3390/app10217741). They reported the association of triglycerides with MetS, a feature not detected by any of the ML algorithms employed in the manuscript. This study may also invalidate the authors' claim regarding the comparability of the manuscript's findings with other published studies on the fact of race. Could the authors elaborate and discuss their findings with the results presented in the link above?

Minor comments

1. It is not clear why the authors used Elliot sigmoid function for WC, SBP, DBP, and why they used the difference between sbp and dbp in the denominator?

2. The CLBE synthetic feature is not mentioned in Table 7 or any other table.

3. The female WC value of 0.45 in the example was calculated based on the male diagnostic criteria of 9 rather than 8.5. The WC value should be 0.66.

4. The authors should share their source code in a public repository for reproducibility and for the benefit of the scientific community.

Reviewer #2: Thank you for offering me the opportunity to review the manuscript “Machine leaning-based predictive model for prevention of metabolic syndrome.” Although, there are a number of publications which have used of machine learning in metabolic syndrome prediction, utilising the data is from other parts of the world, these models might not be relevant in the Korean context due to demographic, cultural and genetic differences. This manuscript uses s significantly larger data set with a broader feature set, applying machine learning approaches to report highly influential features and use them to distinguish cases and control correctly.

- The authors have acquired the medical histories of individuals. What point in time was considered for the development of the model? Was it the latest?

- Samples were collected between 2004 and 2013. However, the analysis was conducted in 2022. Why were the post 2014 samples excluded from the analysis as you highlighted the increase in cases in 2018? If possible, it would be interesting to see model performance on post-2013 data that was trained on pre-2013 data.

- Did authors attempt to test performance for different combinations of feature from different group of features (anthropometric + survey-based features (given), survey-based features +synthetic features and anthropometric + synthetic features) at in round two in Figure 3?

- If possible, please provide a sample distribution by year for both groups.

- If possible, please put your code on GitHub. It is useful to test and reusethe code.

- Is the data available in the public domain, if yes, please provide the link.

- If the list of major cities is not big, could you please list them.

Other corrections:

- Line 10-11: high blood sugar, and dyslipidemia and can lead to cardiovascular disease or type 2 diabetes > high blood sugar and dyslipidemia, and can lead to cardiovascular disease or type 2 diabetes

- Line 31: This tool should be used to predict MetS using only noninvasive information > This tool used to predict MetS using only noninvasive information

- Line 38: The number of features used in the final model, counted based on unsynthesized features. > Only unsynthesized features were counted in the final model.

- Line 59: focusing on classification performance> focusing only on classification performance

- 62: when we try to apply predictive models from previous studies to real life > when we apply predictive models from previous studies to real life.

- 70-71: and compared their usability. How do you compare usability? I guess you mean different parameters such as AUC, accuracy, recall etc.

- 83: All the features were arranged in tabular datasets, excluding outliers and missing values. Did you mean subject instead of feature? If not, how did you identify outlier features?

6. PLOS authors have the option to publish the peer review history of their article (what does this mean?). If published, this will include your full peer review and any attached files.

Reviewer #1: No

Reviewer #2: No

---

## [Author Response · Author response to Decision Letter 0]

21 Mar 2023

We sincerely answered the reviews and uploaded them as a file(Response_to_reviewers.docx).

We reply to the journal requirements as follows.

Journal Requirements:

Answer: We adjusted the format based on the guide.

Answer: In the Methods section, we have provided additional details regarding the ethical approval obtained for the study. 

"The Institutional Review Board (IRB) of Dankook University granted approval for the study protocol and waived the requirement for obtaining informed consent from participants (DKU 2021-06-008)."

Answer: We released the experimental codes in the following online repository.

- https://github.com/shinhseok/predictive-model-for-prevention-of-MetS

4. Please remove any funding-related text from the manuscript and let us know how you would like to update your Funding Statement. 

Answer: We removed the following from the paper and added it to the funding information in the submit system.

"The data used in this study were obtained from the Korean Genome and Epidemiology Study (KoGES; 4851-302), National Institute of Health, Korea Disease Control and Prevention Agency, Republic of Korea."

---

## [Decision Letter · Decision Letter 1]

2 May 2023

PONE-D-23-00127R1Machine learning-based predictive model for prevention of metabolic syndromePLOS ONE

Dear Dr. Shin,

Thank you for submitting your manuscript to PLOS ONE. After careful consideration, we feel that it has merit but does not fully meet PLOS ONE’s publication criteria as it currently stands. Therefore, we invite you to submit a revised version of the manuscript that addresses the points raised during the review process.

I would like to thank the author for having addressed the initial comments and now the manuscript is clearer. There are just a few minor issues that need to be addressed, as raised by one of the reviewers.

The MetS risk map in Fig. 9 showed two boundaries, left diagonal line (WC + BP) and right curve (WC * PB), for patients at risk of MetS. This dividing line and curve were drawn on the basis of the synthetic variables WC and BP. Although the author gave values for WC and BP indicating boundary crossing in their discussion, it would be helpful to provide raw values of waist circumference, systolic blood pressure, diastolic blood pressure and sex for the boundary line and the curve and then compare the raw values of wc, sbp, dbp, and sex to the Korean diagnostic criteria for MetS that s/he presented in the "Preprocessing and Feature Summary" section. This should give an indication of how far or close the raw male/female wc are to risk factors for wc (>=90cm/>=85cm).The script used to generate the risk maps if provided in a shared github repository would be helpful to readers.In the Discussion section the authors should provide an explanation for why high triglycerides and reduced HDL cholesterol were not selected by any of the algorithms though they are potential risk factors for MetS, maybe that these are not the case for Korean population???Please correct all typographical errors including line 76, “of our” Please correct on Line 100, 101, eight cities and eight provinces were provided, not nine. ==============================

We look forward to receiving your revised manuscript.

Kind regards,

Shakuntala Baichoo, Ph.D

Academic Editor

PLOS ONE

Journal Requirements:

Additional Editor Comments:

I would like to thank the author for having addressed the initial comments and now the manuscript is clearer. There are just a few minor issues that need to be addressed, as raised by one of the reviewers.

1. The MetS risk map in Fig. 9 showed two boundaries, left diagonal line (WC + BP) and right curve (WC * PB), for patients at risk of MetS. This dividing line and curve were drawn on the basis of the synthetic variables WC and BP. Although the author gave values for WC and BP indicating boundary crossing in their discussion, it would be helpful to provide raw values of waist circumference, systolic blood pressure, diastolic blood pressure and sex for the boundary line and the curve and then compare the raw values of wc, sbp, dbp, and sex to the Korean diagnostic criteria for MetS that s/he presented in the "Preprocessing and Feature Summary" section. This should give an indication of how far or close the raw male/female wc are to risk factors for wc (>=90cm/>=85cm).

2. The script used to generate the risk maps if provided in a shared github repository would be helpful to readers.

3. In the Discussion section the authors should provide an explanation for why high triglycerides and reduced HDL cholesterol were not selected by any of the algorithms though they are potential risk factors for MetS - maybe that these are not the case for Korean population???

4. Please correct all typographical errors including line 76, “of our”

5. Please correct on Line 100, 101, eight cities and eight provinces were provided, not nine.

Reviewers' comments:

Reviewer's Responses to Questions

**Comments to the Author**

1. If the authors have adequately addressed your comments raised in a previous round of review and you feel that this manuscript is now acceptable for publication, you may indicate that here to bypass the “Comments to the Author” section, enter your conflict of interest statement in the “Confidential to Editor” section, and submit your "Accept" recommendation.

Reviewer #1: (No Response)

Reviewer #3: All comments have been addressed

2. Is the manuscript technically sound, and do the data support the conclusions?

Reviewer #1: Yes

Reviewer #3: Yes

3. Has the statistical analysis been performed appropriately and rigorously? 

Reviewer #1: Yes

Reviewer #3: Yes

4. Have the authors made all data underlying the findings in their manuscript fully available?

Reviewer #1: Yes

Reviewer #3: Yes

5. Is the manuscript presented in an intelligible fashion and written in standard English?

Reviewer #1: Yes

Reviewer #3: No

6. Review Comments to the Author

Reviewer #1: (No Response)

Reviewer #3: Manuscript #: PONE-D-23-00127R1

Title: Machine learning-based predictive model for prevention of metabolic syndrome

Authors: Hyunseok Shin; Simon Shim; Sejong Oh

Article type: Research Article

Authors have addressed reviewers’ comments.

More details have now been provided on participant consent

The procedure is clear.

The preprocessing techniques are details

Sufficient information on parameter fine-tuning

Overall good and satisfactory.

7. PLOS authors have the option to publish the peer review history of their article (what does this mean?). If published, this will include your full peer review and any attached files.

Reviewer #1: No

Reviewer #3: No

---

## [Author Response · Author response to Decision Letter 1]

10 May 2023

We have thoroughly addressed all of the comments and suggestions provided by the reviewers and have provided our responses in the "Response_to_reviewers" file.

---

## [Editor Report · Decision Letter 2]

22 May 2023

Machine learning-based predictive model for prevention of metabolic syndrome

PONE-D-23-00127R2

Dear Dr. Shin,

We’re pleased to inform you that your manuscript has been judged scientifically suitable for publication and will be formally accepted for publication once it meets all outstanding technical requirements.

Kind regards,

Shakuntala Baichoo, Ph.D

Academic Editor

PLOS ONE

Additional Editor Comments (optional):

Thanks for addressing all the comments.
---

## [Editor Report · Acceptance letter]

24 May 2023

PONE-D-23-00127R2 

Machine learning-based predictive model for prevention of metabolic syndrome 

Dear Dr. Shin:

I'm pleased to inform you that your manuscript has been deemed suitable for publication in PLOS ONE. Congratulations! Your manuscript is now with our production department. 

Kind regards, 

on behalf of

Dr. Shakuntala Baichoo 

Academic Editor

PLOS ONE